# Gate to the Vessel: Residual Experts Restore What SAM Overlooks

**Weili Jiang**
School of Computing and Artificial Intelligence
Southwest Jiaotong University
jiangweili@swjtu.edu.cn

**Jinrong Lv**
School of Computing and Artificial Intelligence
Southwest Jiaotong University
lvjinrong@my.swjtu.edu.cn

**Xun Gong**
School of Computing and Artificial Intelligence
Southwest Jiaotong University
xgong@swjtu.edu.cn

**Xiaomeng Li**
Department of Electronic and Computer Engineering
The Hong Kong University of Science and Technology
eexmli@ust.hk

**Chubin Ou**[*]
Institute of Biomedical Engineering, Peking University Shenzhen Graduate School
Department of Radiology, Guangdong Provincial People's Hospital
cou@connect.ust.hk

## Abstract

Foundation segmentation models like Segment Anything (SAM) exhibit strong generalization on natural images but struggle with localized failures in medical imaging, especially on fine-grained structures such as vessels with complex morphology and indistinct boundaries. To address this, we propose FineSAM++, a structure-aware sparse expert framework designed to refine SAM outputs by introducing a confidence-driven soft Routing Module. This module dynamically identifies structurally uncertain regions and activates a lightweight Residual Expert to model and correct residual structural errors only within these areas, thereby achieving efficient "refinement over retraining." Extensive experiments on five public vascular segmentation datasets demonstrate that FineSAM++ consistently outperforms both SAM-adapted baselines and task-specific models in terms of accuracy, topological consistency. Our results highlight the effectiveness of sparse, structure-driven Mixture-of-Experts (MoE) strategies for enhancing the reliability of foundation vision models in clinical image understanding tasks.

---

[*]Corresponding author.

39th Conference on Neural Information Processing Systems (NeurIPS 2025).

# 1 Introduction

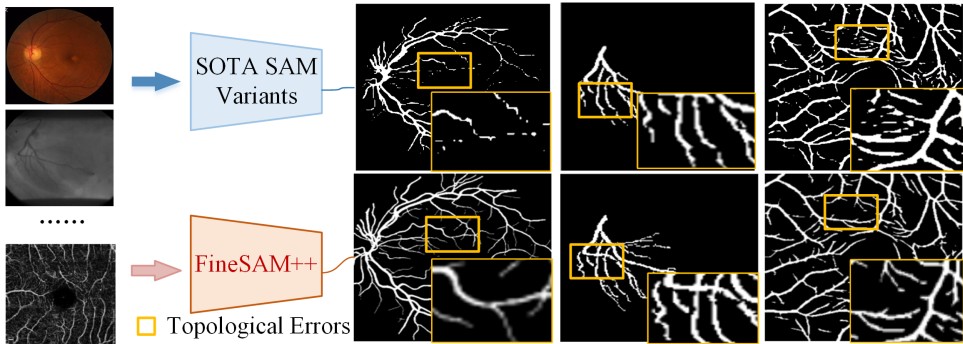

Figure 1: Illustration of the core motivation behind FineSAM++.

Foundation vision models such as Segment Anything Model (SAM)[1] have demonstrated remarkable prompt responsiveness and zero-shot generalization capabilities for natural image segmentation tasks. With the increasing popularity of SAM in general-purpose computer vision, recent efforts have explored its adaptation to medical image segmentation [2–4]. However, empirical studies consistently report substantial performance degradation when SAM is applied to medical structures, especially for fine-grained targets like vessels, where predictions often suffer from topological errors, including disconnections, boundary ambiguity, and local omissions [4, 5] (see Fig.1).

To bridge the domain gap between natural and medical images, existing approaches have explored domain adaptation strategies including Adapters [6, 7], LoRA [8], prompt generation [9], and SAM-CLIP hybrid models [10, 11] (see Fig.2). Nevertheless, these methods primarily focus on aligning global semantic representations, leaving structurally ambiguous or uncertain regions under-modeled [5, 4]. As a result, they fail to resolve the persistent issue of local structural degradation. We further observe that mispredictions in medical images predominantly occur around blurred boundaries or fine structural details, which typically manifest as high uncertainty or large residual deviations from the ground truth. This suggests that a unified global adaptation scheme inherently struggles to satisfy both semantic alignment and structural recovery, as the modeling objectives present a natural conflict.

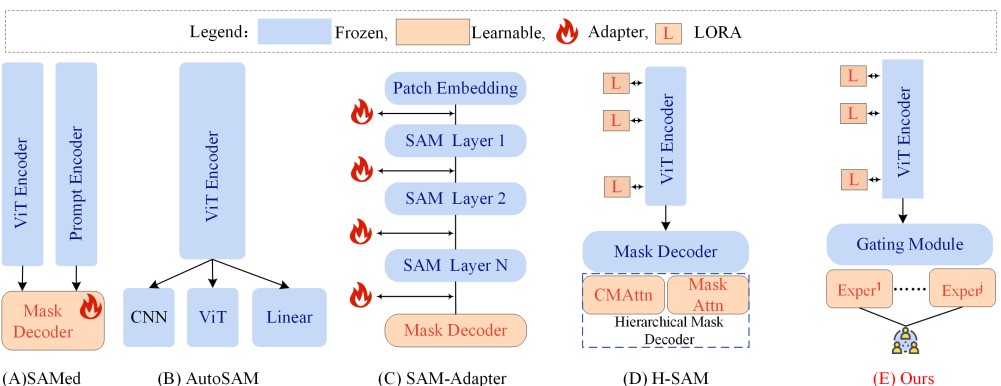

Figure 2: Comparison of SAM-based adaptation methods for medical image segmentation.

Inspired by the sparse activation of Mixture-of-Experts (MoE) architectures in large model design [12–14], we argue that MoE provides a promising paradigm for fine-grained structure modeling. By dynamically activating expert pathways only where necessary, an MoE-inspired framework can focus modeling capacity on local difficult regions without sacrificing overall efficiency. Based on this principle, we propose **FineSAM++**, a structure-enhanced framework following the MoE philosophy. FineSAM++ introduces two expert modules: a global LoRA Expert for domain adaptation and a Residual Expert for local residual correction and structural refinement. Their activation and

cooperation are jointly controlled by an uncertainty-aware Gating Module, resulting in an efficient *shared-backbone + localized refinement* strategy. Our main contributions are summarized as follows.

- We propose **FineSAM++**, the structure-aware sparse MoE framework that integrates multiple localized residual correction pathways into frozen foundation segmentation models like SAM, addressing their systematic failures on fine-grained medical structures.

- We design a **spatially-aware soft routing mechanism** that jointly predicts spatial uncertainty masks and fractional expert routing weights, dynamically activating only a small subset of residual experts for structurally ambiguous regions.

- We conduct extensive experiments to validate the effectiveness of the proposed module and achieve state-of-the-art performance on five public vessel datasets covering three distinct imaging modalities.

## 2 Related Work

### 2.1 Foundation Models for Medical Image Segmentation

Foundation vision models such as the SAM [1] have demonstrated strong generalization and zero-shot capabilities in natural image segmentation. However, their performance degrades significantly in medical imaging tasks, particularly for fine-grained structures like vessels and retinal layers, due to domain shifts and a lack of priors for thin, low-contrast anatomy[15, 16]. To bridge this gap, recent studies have explored adapter-based tuning [6, 7], LoRA-based silent fine-tuning [8], automatic prompt generation [9], and hybrid approaches combining SAM with CLIP [10, 11], as well as methods that improve boundary accuracy through high-quality priors and edge-aware refinement [17]. While these methods improve global adaptability, they often overlook localized structural failures—such as vessel discontinuities, blurred edges, and fragmented predictions—that are critical in clinical applications [18]. FineSAM++ addresses this limitation by introducing a sparse residual expert framework guided by uncertainty-aware gating, enabling selective correction of structurally uncertain regions while preserving global semantic consistency.

### 2.2 Mixture-of-Experts Architectures in Vision Modeling

Mixture-of-Experts (MoE) architectures have emerged as a powerful paradigm for scaling deep networks while maintaining efficiency [12, 13]. In vision, recent works have explored various MoE formulations for different purposes. Switch Transformers [19] propose token-based routing to conditionally activate expert blocks in large-scale transformers. Expert Choice Routing [14] and SwitchHead [20] further improve routing efficiency and stability by optimizing expert selection and assignment. CuMo [21] introduces co-upcycled expert reuse to scale multimodal models, achieving strong performance with limited expert redundancy. Neural Experts [22] and related vision-specific MoE designs have focused primarily on large-scale classification and vision-language pretraining tasks. However, these works focus on token or patch-level routing for classification or vision-language tasks, and have not explored dense prediction or fine-grained structural correction. FineSAM++ to systematically integrate sparse expert routing into a dense segmentation pipeline. By introducing a soft Gating Module and Residual Expert, FineSAM++ applies MoE principles to address local structural inconsistencies in medical vessel segmentation, which remains largely underexplored in prior vision MoE literature.

## 3 Method

Inspired by the success of sparse MoE in scaling LLMs and vision models [12, 13], we propose FineSAM++, a sparse expert framework designed for fine-grained targets. Fine-grained target segmentation naturally fits the MoE paradigm due to the extreme sparsity, topology irregularity, and strong locality of error-prone regions. Our framework incorporates two specialized lightweight experts: a global LoRA Expert for domain adaptation and a Residual Expert for structure-aware local residual correction. A differentiable Gating Module inspired by Expert Choice Routing [21] coordinates dynamic activation of experts.

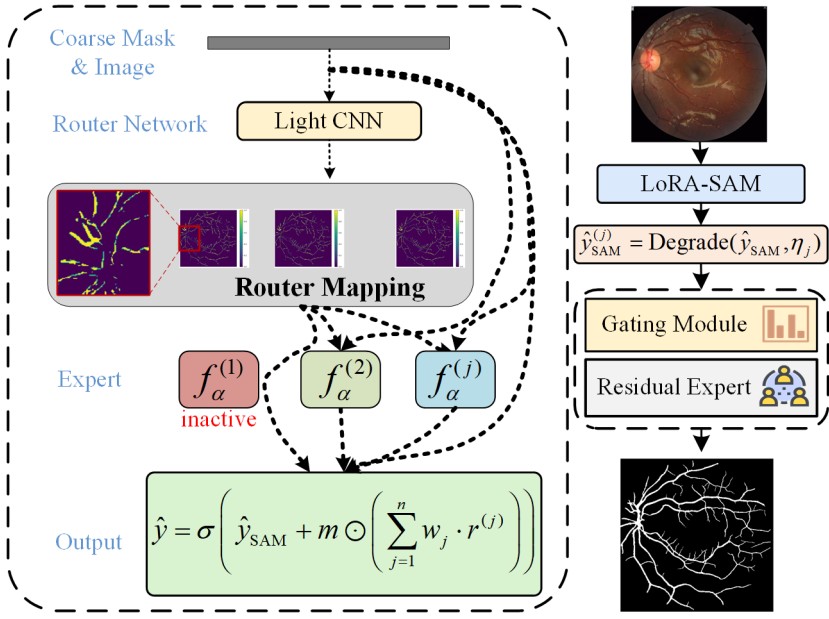

Figure 3: Overview of the FineSAM++ architecture.

## 3.1 Overview

Given an input image $x \in \mathbb{R}^{H \times W \times 3}$, FineSAM++ first generates a coarse segmentation prediction $\hat{y}_{\text{SAM}} \in [0,1]^{H \times W}$ using a LoRA-SAM model with degraded inputs (see Sec. 3.3 for details). Here, LoRA-SAM refers to a frozen SAM backbone augmented with LoRA adapters, following the parameter-efficient fine-tuning strategy in [2]. Additionally, for the prompt encoder in LoRA-SAM, FineSAM++ does not require any manual prompts; instead, it learns a fixed default embedding during training. While the coarse prediction is generally effective, it often exhibits topological inconsistencies in challenging regions such as vascular bifurcations, blurred boundaries, and disconnected thin structures.

To address this, FineSAM++ introduces a modular sparse refinement pathway consisting of (1) a Gating Module to estimate spatial uncertainty and expert routing weights, and (2) $J$ lightweight Residual Experts to perform localized residual correction (see Fig.3). Each Residual Expert receives a perturbed variant of the coarse mask to promote specialization. The final output is obtained by fusing the original SAM prediction with the aggregated residual corrections:

$$\hat{y} = \sigma \left( \hat{y}_{\text{SAM}} + m \odot \left( \sum_{j=1}^{J} w_j \cdot r^{(j)} \right) \right), \tag{1}$$

where $m$ is the uncertainty mask, $w_j$ are routing weights, and $r^{(j)}$ are the expert outputs. $\odot$ denotes element-wise multiplication.

## 3.2 Gating Module

Classical MoE architectures rely on token-level routing based on dense embedding vectors [12, 19]. However, dense vision transformers like SAM produce structured 2D feature maps, where localized spatial uncertainty plays a critical role. To address this gap, FineSAM++ introduces a **spatially-aware soft routing mechanism** via a dedicated Gating module. Our Gating module $g_\theta$ serves two purposes: (1) generate a soft mask $m \in [0,1]^{H \times W}$ indicating spatial uncertainty at each pixel, and (2) output fractional routing weights $\{w_j\}$ for $J$ parallel Residual Experts. The module receives the concatenation of the image $x$ and coarse mask $\hat{y}_{\text{SAM}}$ as input, capturing both appearance and prediction context:

$$m, \{w_j\} = g_\theta(\text{Concat}(x, \hat{y}_{\text{SAM}})). \tag{2}$$

This design is fundamentally different from standard MoE routers, which treat each input as independent. Instead, our Gating module explicitly leverages spatial correlations, identifying localized regions that require residual correction. By assigning soft weights to multiple experts, the routing mechanism enables fine-grained specialization without hard top-k decisions, which are known to suffer from instability and expert imbalance [14, 22].

We supervise the Gating module using pseudo-labels derived from backbone prediction errors. A binary pseudo-label mask $g_t$ is first generated by thresholding the absolute error between the SAM coarse prediction $\hat{y}_{\text{SAM}}$ and the ground truth label $y$:

$$g_t(i) = \mathbb{I}\left(|\hat{y}_{\text{SAM}}(i) - y(i)| > \delta\right), \tag{3}$$

where $\mathbb{I}(\cdot)$ is the indicator function that outputs 1 if the condition holds and 0 otherwise, and $\delta$ is a pre-defined error threshold. The Gating module is trained to predict a router map $m \in [0,1]^{H \times W}$, where higher values indicate greater structural uncertainty. We optimize the Gating output using the standard binary cross-entropy (BCE) loss:

$$\mathcal{L}_{\text{gate}} = -\frac{1}{N}\sum_i \left[g_t(i)\log m(i) + (1 - g_t(i))\log(1 - m(i))\right], \tag{4}$$

where $N$ is the total number of pixels. This loss encourages the Gating module to activate only in regions where the backbone prediction is structurally unreliable, while suppressing unnecessary expert invocation elsewhere.

### 3.3 Residual Experts

Conventional segmentation refinement frameworks either retrain the full model or introduce a single auxiliary head for residual prediction [8, 2]. In contrast, FineSAM++ proposes a **diverse multi-expert residual correction scheme**. Inspired by MoE principles of expert specialization, we deploy $J$ parallel Residual Experts $f_\alpha^{(j)}$, each receiving a slightly degraded version of the coarse mask:

$$\hat{y}_{\text{SAM}}^{(j)} = \text{Degrade}(\hat{y}_{\text{SAM}}, \eta_j), \tag{5}$$

where $\text{Degrade}(\cdot)$ applies random masking, noise injection, or occlusion perturbations to promote input diversity. This novel perturbation-based expert diversification allows each expert to specialize in correcting specific structural failures, such as disconnections, thin vessel loss, or noisy edges.

Each expert predicts residual corrections conditioned on both the perturbed coarse mask $\hat{y}_{\text{SAM}}^{(j)}$ and input image $x$:

$$r^{(j)} = f_\alpha^{(j)}(x, \hat{y}_{\text{SAM}}^{(j)}), \tag{6}$$

where $r^{(j)} \in \mathbb{R}^{H \times W}$ denotes the residual correction map predicted by the $j$-th expert. This formulation allows each expert to focus on different perturbation patterns and promotes specialization. Unlike prior works that produce complete segmentation masks, our experts focus exclusively on **local residual correction**. We supervise this process using a gated *mean squared error (MSE)* loss between the fused expert output and the ground truth:

$$\mathcal{L}_{\text{res}} = \frac{1}{N}\sum_i \left( (m \cdot \sum_{j=1}^{J} w_j \cdot r^{(j)})(i) + (\hat{y}_{\text{SAM}}(i) - y(i)) \right)^2, \tag{7}$$

where $N$ is the total number of pixels, $m(i) \in [0,1]$ is the soft spatial uncertainty mask from the Gating Module at pixel $i$, $w_j$ are the normalized routing weights for each expert, and $y(i)$ is the ground truth label.

### 3.4 Progressive Optimization with Dynamic Weighting

**Training strategy with two-phase adaptive learning.** The Residual Expert strongly depends on stable backbone predictions to avoid overfitting to noisy residual targets. Therefore, we design a two-phase adaptive training strategy. In Phase 1, we freeze both the Gating Module and Residual Expert and train only the SAM backbone until $\mathcal{L}_{\text{SAM}}$ falls below a threshold $\epsilon$ for $k$ consecutive epochs. This warm-up stage stabilizes the coarse prediction output.

Once convergence is detected, Phase 2 activates the residual correction pathway and applies dynamic loss weighting $\lambda_{\text{res}}(t)$:

$$\lambda_{\text{res}}(t) = \min\left(1, \frac{t - t_0}{T}\right), \quad \lambda_{\text{SAM}}(t) = 1 - \lambda_{\text{res}}(t), \tag{8}$$

where $t_0$ is the warm-up completion epoch and $T$ controls the progressive ramp-up of the Residual Expert contribution. This two-phase curriculum minimizes early instability and ensures smoother joint optimization of both the backbone and expert modules.

**Overall Loss.** The final training objective of FineSAM++ combines the semantic segmentation loss of the SAM backbone and the structural refinement losses of the Residual Expert:

$$\mathcal{L} = \lambda_{\text{SAM}}\mathcal{L}_{\text{SAM}} + \lambda_{\text{res}}\mathcal{L}_{\text{res}} + \lambda_{\text{gate}}\mathcal{L}_{\text{gate}}, \tag{9}$$

where $\mathcal{L}_{\text{SAM}}$ represents the combined Dice and binary cross-entropy loss on the backbone output, $\mathcal{L}_{\text{res}}$ is the masked residual regression loss, and $\mathcal{L}_{\text{gate}}$ is the gating supervision loss. The weighting coefficients $\lambda$ balance the contributions of each component and are dynamically adjusted as described above. This formulation enables FineSAM++ to jointly optimize global semantic consistency and local structural refinement in a stable and interpretable manner.

Table 1: Quantitative comparison on DRIVE, DCAI, CHUAC and ROSE datasets. The best results are bolded while the second best are underlined. **Other Dataset (FIVES) quantitative comparison are provided in the supplementary material.**

| Data | Method | Dice | ACC | AUC | SE | SP | Data | Method | Dice | ACC | AUC | SE | SP |
|---|---|---|---|---|---|---|---|---|---|---|---|---|---|
| DRIVE | U-Net | 0.7787 | 0.9616 | 0.9863 | 0.7802 | 0.9792 | DCAI | U-Net | 0.7392 | 0.9741 | 0.9803 | 0.7647 | 0.9851 |
| | Att U-Net | 0.7808 | 0.9621 | 0.9774 | 0.7931 | 0.9795 | | Att U-Net | 0.7511 | 0.9753 | 0.9834 | 0.7851 | 0.9861 |
| | U-Net++ | 0.7860 | 0.9635 | 0.9825 | 0.7891 | 0.9850 | | U-Net++ | 0.7766 | 0.9757 | 0.9860 | 0.7932 | 0.9857 |
| | R2U-Net | 0.8171 | 0.9556 | 0.9784 | 0.7792 | 0.9813 | | CS-Net | 0.7790 | 0.9763 | 0.9889 | 0.7895 | 0.9813 |
| | TransUNet | 0.7872 | 0.9577 | 0.9792 | 0.7819 | 0.9788 | | VSSC Net | - | 0.9700 | 0.9831 | 0.7728 | 0.9809 |
| | CAViT | - | 0.9700 | 0.9864 | 0.7924 | 0.9872 | | FR-UNet | 0.7736 | 0.9744 | 0.9897 | 0.8344 | 0.9824 |
| | MCDAU-Net | 0.8129 | 0.9589 | - | 0.8215 | 0.9739 | | MedUNAS | 0.7820 | **0.9800** | - | 0.8089 | 0.9905 |
| | Retina-TransNet | 0.7964 | - | 0.8836 | 0.7850 | 0.9821 | | G2ViT | 0.7659 | 0.9761 | 0.9904 | 0.8387 | **0.9914** |
| | MRC-Net | - | **0.9698** | 0.9825 | 0.8250 | 0.9837 | | HRNet | 0.7919 | 0.9777 | 0.9899 | 0.8007 | 0.9876 |
| | Gupta et al | 0.7978 | 0.9677 | 0.8843 | 0.7863 | 0.9824 | | Gupta et al | 0.7938 | 0.9681 | 0.9911 | 0.8853 | 0.9891 |
| | RETFound | 0.8020 | 0.9649 | 0.8830 | 0.7796 | 0.9821 | | RETFound | 0.7948 | 0.9685 | 0.9923 | 0.8857 | 0.9872 |
| | nnUnet | 0.8220 | 0.9698 | 0.8940 | 0.8019 | 0.9862 | | nnUnet | 0.8045 | 0.9584 | 0.9903 | 0.8264 | 0.9879 |
| | SAM Aapter | 0.4498 | 0.9311 | 0.9204 | 0.7577 | 0.9377 | | SAM Aapter | 0.7583 | 0.9727 | 0.9408 | 0.7882 | 0.9836 |
| | H-SAM | 0.6622 | 0.9485 | 0.7824 | 0.5808 | **0.9840** | | H-SAM | 0.6374 | 0.9661 | 0.7810 | 0.5732 | 0.9887 |
| | AutoSAm | 0.6603 | 0.9414 | **0.9872** | **0.8368** | 0.9822 | | AutoSAm | 0.7175 | 0.9693 | 0.8483 | 0.7120 | 0.9760 |
| | SAMed | 0.6170 | 0.9450 | 0.9600 | 0.5070 | 0.9880 | | SAMed | 0.5750 | 0.9540 | 0.9550 | 0.5750 | 0.9760 |
| | HQ-SAM | 0.7978 | 0.9697 | 0.8824 | 0.8033 | 0.9824 | | HQ-SAM | 0.7880 | 0.9770 | 0.8890 | 0.7890 | 0.988 |
| | **Ours** | **0.8231** | 0.9790 | 0.9870 | 0.8366 | 0.9834 | | **Ours** | **0.8127** | 0.9775 | **0.9931** | **0.8479** | 0.9872 |
| CHUAC | U-Net | 0.6768 | 0.9744 | 0.9582 | 0.5801 | 0.9941 | ROSE | U-Net | 0.7116 | 0.8955 | 0.9218 | 0.7867 | 0.8780 |
| | Att U-Net | 0.6941 | 0.9803 | 0.9515 | 0.6420 | 0.9922 | | CS-Net | 0.7608 | 0.9152 | 0.9392 | 0.8631 | 0.9112 |
| | U-Net++ | 0.7000 | 0.9802 | 0.9669 | 0.6109 | **0.9949** | | CE-Net | 0.7511 | 0.9121 | 0.9292 | - | - |
| | CS-Net | 0.7171 | 0.9796 | 0.9747 | 0.6735 | 0.9918 | | COSFIRE | 0.7517 | 0.9227 | 0.9286 | - | - |
| | VSSC Net | - | 0.9721 | 0.9757 | 0.7892 | 0.9797 | | COOF | 0.6606 | 0.8530 | 0.8689 | - | - |
| | FR-UNet | 0.7543 | 0.9740 | 0.9786 | 0.7836 | 0.9867 | | ResU-Net | 0.7461 | 0.9098 | 0.9252 | - | - |
| | MedUNAS | 0.7456 | 0.9807 | - | 0.7829 | 0.9912 | | DUNet | 0.7505 | 0.9118 | 0.9334 | - | - |
| | G2ViT | 0.7612 | 0.9809 | 0.9858 | 0.7908 | 0.9950 | | three-stage | 0.7663 | 0.9179 | 0.9179 | - | - |
| | HRNet | 0.7526 | 0.9811 | 0.9906 | 0.7456 | 0.9906 | | OCTA-Net | 0.7697 | 0.9182 | 0.9453 | - | - |
| | Gupta et al | 0.7168 | 0.9799 | 0.9739 | 0.6728 | 0.9907 | | Gupta et al | 0.7601 | 0.9164 | 0.9399 | 0.8563 | 0.9109 |
| | RETFound | 0.7636 | 0.9604 | 0.9904 | 0.7325 | 0.9906 | | RETFound | 0.7126 | 0.9197 | 0.9337 | 0.8563 | 0.9193 |
| | nnUnet | **0.7814** | 0.9776 | 0.8842 | 0.7788 | 0.9896 | | nnUnet | 0.8270 | 0.9470 | 0.9310 | 0.8650 | **0.9940** |
| | SAM Aapter | 0.7636 | 0.9784 | 0.9359 | 0.7583 | 0.9902 | | SAM Aapter | 0.6316 | 0.8578 | 0.8451 | 0.6503 | 0.9801 |
| | H-SAM | 0.6951 | 0.9707 | 0.8310 | 0.6758 | 0.9862 | | H-SAM | 0.6968 | 0.8973 | 0.7965 | 0.6335 | 0.9595 |
| | AutoSAm | 0.6833 | 0.9654 | 0.8614 | 0.7457 | 0.9772 | | AutoSAM | 0.6954 | 0.8949 | 0.8054 | 0.6557 | 0.9684 |
| | SAMed | 0.7520 | 0.9790 | 0.9880 | 0.7040 | 0.9920 | | SAMed | 0.6390 | 0.8810 | 0.8830 | 0.5600 | 0.9570 |
| | HQ-SAM | 0.7050 | 0.8940 | 0.8120 | 0.6760 | 0.9480 | | HQ-SAM | 0.7520 | **0.9609** | textbf0.9904 | 0.7940 | 0.9887 |
| | **Ours** | 0.7768 | 0.9807 | **0.9951** | 0.7567 | 0.9932 | | **Ours** | **0.8220** | 0.9483 | 0.9827 | **0.9485** | 0.9823 |

## 4 Experiments

### 4.1 Implementation Details

**Dataset.** We evaluate FineSAM++ across five publicly available vascular segmentation datasets spanning three imaging modalities. The DRIVE dataset [23] contains two-dimensional retinal fundus images with ground truth vessel masks. ROSE [24] provides retinal vessel segmentation from 2D optical coherence tomography angiography (OCTA) scans. FIVES [25] includes 800 high-resolution multi-disease color fundus photographs annotated for vessel structures. DCA1 [26] and CHUAC [27] are coronary angiography datasets containing fluoroscopic X-ray vessel images. We select these datasets to cover the full spectrum of challenges targeted by FineSAM++, including variations in

anatomical regions (retina vs. coronary arteries), imaging modalities (fundus photography, OCTA, X-ray angiography), and segmentation difficulties (low contrast, thin structures, fragmented vessels). **Detailed dataset statistics and preprocessing steps are provided in the supplementary material.**

**Beyond vascular segmentation.** To further assess cross-domain generalization, we additionally evaluate multi-class abdominal organ segmentation on the **Synapse Multi-Organ CT** dataset (eight organs), demonstrating that FineSAM++ maintains strong performance outside the vascular domain. **Detailed dataset statistics and results are provided in the supplementary material.**

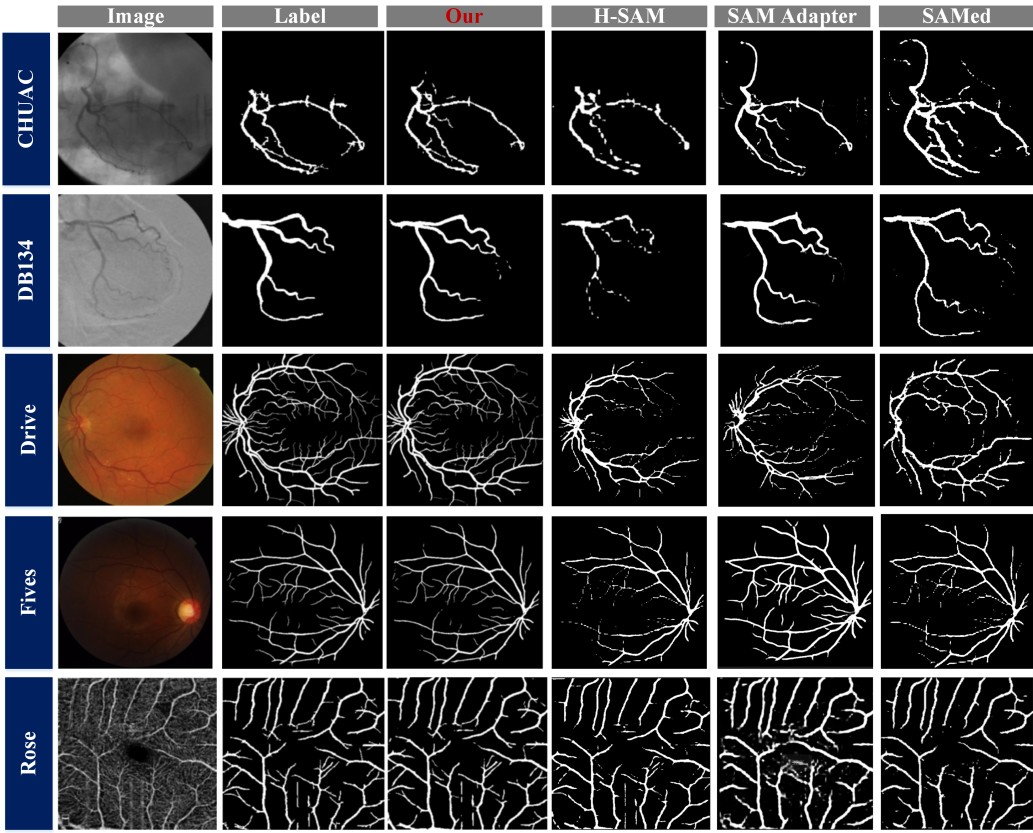

Figure 4: Qualitative comparison of FineSAM++ against H-SAM, SAM Adapter, and SAMed across five datasets. FineSAM++ provides more continuous and complete vessel segmentation with fewer false positives and fragmentation artifacts.

**Training settings.** All experiments are implemented using PyTorch and trained on two NVIDIA RTX 4090 GPUs. Data augmentation includes random elastic deformation, rotation, scaling, and intensity jittering. For the backbone, we follow [2] and integrate LoRA adapters into the frozen SAM encoder with a rank of 4. We adopt the ViT-B configuration of SAM as the base encoder. For fair comparison across datasets, all images are resized to $512 \times 512$ resolution. The maximum training epoch is set to 300. We use the AdamW optimizer with $\beta_1 = 0.9$, $\beta_2 = 0.999$, and weight decay of 0.1. The initial learning rate is set to $5 \times 10^{-5}$ and decayed using a cosine annealing schedule. All hyperparameters are fixed across datasets without additional tuning to ensure fair comparison and reproducibility.

**Evaluation metrics.** To comprehensively assess model performance, we evaluate FineSAM++ and all baselines across standard segmentation accuracy metrics and specialized structural consistency metrics tailored for vascular image analysis. The primary evaluation metrics include Dice, accuracy (ACC), sensitivity (SE), specificity (SP), connectivity (C), overlapping area (A), vessel length consistency (L), and centerline Dice (clDice). **For completeness and reproducibility, detailed definitions for all metrics are provided in the supplementary material.**

**Compared methods.** In this study, we comprehensively benchmark FineSAM++ against a wide range of state-of-the-art (SOTA) methods previously reported on the evaluated datasets. The baselines

include both classical CNN-based and modern Transformer-based segmentation models, topology-aware segmentation method, as well as several recent SAM-variant foundation model adaptations. **For clarity and reproducibility, the full list of compared methods and corresponding references are provided in the supplementary material.**

Table 2: Quantitative comparison on DRIVE, FIVES, DCAI, CHUAC and ROSE datasets. Metrics include connectivity (C), area accuracy (A), length similarity (L), and centerline Dice (ClDice). The best results are bolded and the second best are underlined.

| Data | Method | Metrics | | | | Data | Method | Metrics | | | |
|---|---|---|---|---|---|---|---|---|---|---|---|
| | | C | A | L | ClDice | | | C | A | L | ClDice |
| DRIVE | U-Net | **0.998** | 0.712 | 0.608 | 0.761 | FIVES | U-Net | 0.994 | 0.897 | 0.912 | 0.889 |
| | Att U-Net | 0.994 | 0.768 | 0.732 | 0.775 | | Swin-Unet | 0.997 | 0.863 | 0.871 | 0.785 |
| | U-Net++ | 0.994 | 0.783 | 0.774 | 0.801 | | TransUnet | 0.997 | 0.919 | 0.923 | 0.911 |
| | SAM Adapter | 0.993 | 0.412 | 0.321 | 0.488 | | SAM Adapter | 0.993 | 0.713 | 0.654 | 0.878 |
| | H-SAM | 0.997 | 0.631 | 0.654 | 0.612 | | H-SAM | 0.994 | 0.698 | 0.711 | 0.645 |
| | AutoSAM | **0.998** | 0.597 | 0.621 | 0.598 | | AutoSAM | 0.996 | 0.652 | 0.675 | 0.887 |
| | SAMed | 0.996 | 0.583 | 0.561 | 0.556 | | SAMed | 0.994 | 0.691 | 0.712 | 0.657 |
| | Ours | **0.998** | **0.848** | **0.865** | **0.832** | | Ours | **0.997** | **0.921** | **0.925** | **0.914** |
| DCAI | U-Net | 0.995 | 0.78 | 0.812 | 0.7900 | ROSE | U-Net | 0.996 | 0.723 | 0.739 | 0.7100 |
| | Att U-Net | 0.996 | 0.812 | 0.798 | 0.8050 | | CS-Net | 0.997 | 0.776 | 0.789 | 0.7500 |
| | U-Net++ | **0.998** | 0.831 | 0.813 | 0.8150 | | OCTA-Net | **0.999** | 0.781 | 0.765 | 0.7550 |
| | SAM Adapter | **0.998** | 0.785 | 0.812 | 0.8 | | SAM Adapter | 0.992 | 0.683 | 0.657 | 0.6600 |
| | H-SAM | 0.992 | 0.732 | 0.734 | 0.72 | | H-SAM | 0.994 | 0.721 | 0.719 | 0.7050 |
| | AutoSAM | 0.995 | 0.732 | 0.757 | 0.735 | | AutoSAM | 0.995 | 0.736 | 0.743 | 0.7150 |
| | SAMed | 0.997 | 0.643 | 0.651 | 0.65 | | SAMed | 0.997 | 0.675 | 0.674 | 0.6900 |
| | Ours | 0.997 | **0.903** | **0.877** | **0.865** | | Ours | 0.997 | **0.819** | **0.839** | 0.8050 |
| CHUAC | U-Net | 0.994 | 0.631 | 0.629 | 0.6200 | CHUAC | H-SAM | 0.994 | 0.712 | 0.719 | 0.7000 |
| | Att U-Net | 0.995 | 0.702 | 0.698 | 0.6850 | | AutoSAM | 0.993 | 0.698 | 0.723 | 0.6900 |
| | U-Net++ | 0.996 | 0.723 | 0.722 | 0.7150 | | SAMed | 0.997 | 0.757 | 0.739 | 0.7350 |
| | SAM Adapter | **0.998** | 0.759 | 0.768 | 0.75 | | Ours | **0.998** | **0.787** | **0.795** | **0.7700** |

## 4.2 Main Results

**Quantitative Comparisons.** Tab. 1 summarizes the performance comparison across four datasets. FineSAM++ consistently achieves superior results over both CNN- and Transformer-based baselines as well as recent SAM-derived methods. On DRIVE and DCAI, FineSAM++ sets new state-of-the-art Dice scores of 0.8231 and 0.8127, respectively, substantially outperforming AutoSAM (0.6603 and 0.7175) and H-SAM (0.6622 and 0.6374). On CHUAC and ROSE, our method also delivers the highest Dice scores (0.7768 and 0.8220), demonstrating robust generalization across diverse vascular modalities. These results validate the effectiveness of our multi-expert sparse refinement strategy in addressing localized structural failures of foundation segmentation models while maintaining high global consistency.

**Qualitative Results.** Fig. 4 shows representative qualitative comparisons of FineSAM++ against leading SAM-variant methods (H-SAM, SAM Adapter, SAMed) across five datasets. Our method consistently produces sharper and more continuous vessel structures with fewer false positives and disconnected branches. In coronary angiography datasets (CHUAC, DB134), FineSAM++ better captures thin vessel bifurcations and suppresses background noise. On retinal fundus images (DRIVE, FIVES), our model recovers small peripheral vessels missed by baselines. For OCTA images (ROSE), FineSAM++ yields smoother centerlines with significantly reduced fragmentation compared to prior approaches. These visual improvements highlight the advantage of our multi-expert sparse refinement design for addressing localized structural errors while preserving global topology.

**Topological analysis.** Tab. 7 reports connectivity (C), area (A), length (L), and ClDice metrics across four datasets. FineSAM++ consistently achieves the highest ClDice scores, indicating superior preservation of vessel topology and centerline continuity. On DRIVE and DCAI, our method outperforms the strongest baseline by margins of 0.832 vs. 0.801 and 0.865 vs. 0.815 respectively. Similar trends are observed on CHUAC and ROSE. The strong gains in connectivity (C) and length (L) further highlight the advantage of our sparse expert refinement design in correcting disconnections and fragmented vessels present in the coarse backbone predictions. These results demonstrate that FineSAM++ not only improves segmentation accuracy but also enhances structural fidelity, which is critical in clinical vascular analysis.

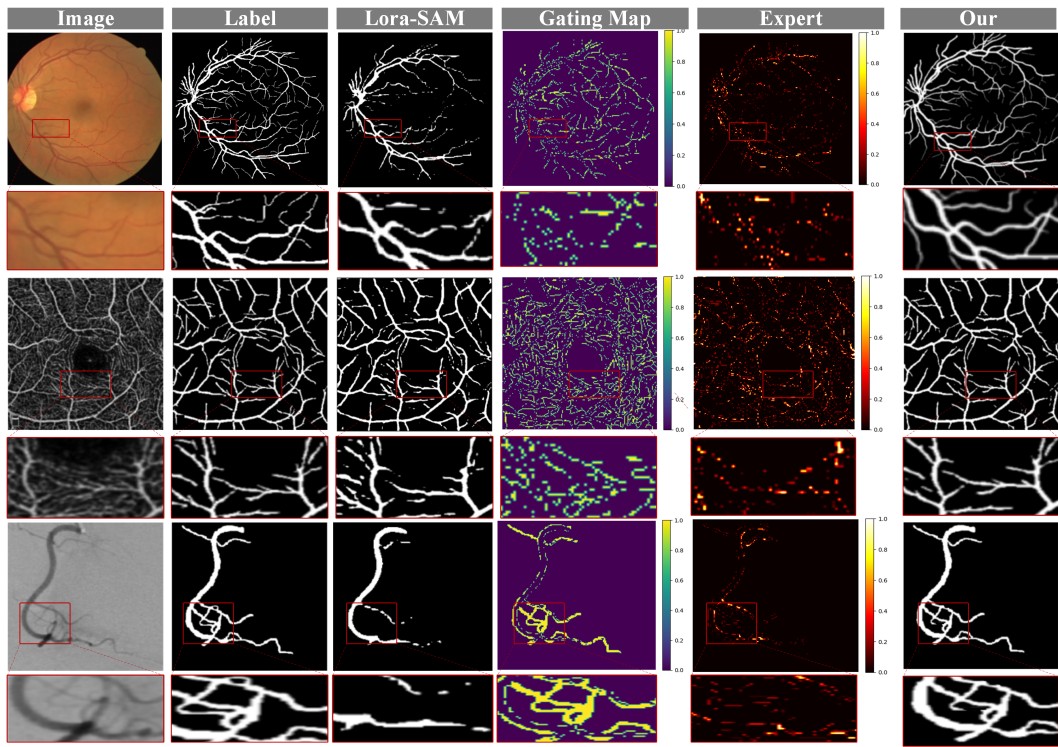

Figure 5: Visualization of the FineSAM++ refinement process.

## 4.3 Ablation Study

**Parameter Efficiency.** To address concerns about the parameter efficiency of our proposed method **FineSAM++** among SAM-based segmentation approaches, we conducted a comprehensive comparison under a standard input resolution of $(1, 3, 1024, 1024)$. Specifically, we measured the number of parameters, FLOPs (GAMCs), and inference latency, as summarized in Table 3. As shown, Fine-SAM++ introduces only about **0.7M** additional learnable parameters over the SAM backbone (94.4M vs. 93.7M). Compared with other SAM-based methods—such as SAM Adapter (104.3M), H-SAM (111.3M), and AutoSAM (135.29M)—FineSAM++ demonstrates substantially higher parameter efficiency. Moreover, while its total parameter count is higher than lightweight architectures like Unet, FineSAM++ achieves the highest Dice score (0.8231) among all evaluated methods. These results indicate that FineSAM++ achieves an excellent balance between parameter efficiency and segmentation performance.

Table 3: Comparison of model size, computational cost, latency, and segmentation accuracy (Dice score) across segmentation methods using an input of size (1, 3, 1024, 1024).

| Model | Params (M) | GAMCs (G) | Latency (ms) | Dice |
|---|---|---|---|---|
| Unets | 34.53 | 4.08 | 1.10 | 0.7787 |
| nnUnet | 126.2 | 1864.9 | 37.4 | 0.8220 |
| SAM Adapter | 104.3 | 400.1 | 127.8 | 0.4498 |
| H-SAM | 111.3 | 370.6 | 124.8 | 0.6622 |
| AutoSAM | 135.29 | 774.16 | 166.22 | 0.6603 |
| SAMed | 92.2 | 370.5 | 117.1 | 0.6170 |
| SAM | 93.7 | 372.0 | 116.33 | / |
| Ours (FineSAM++) | 94.4 | 376.8 | 117.6 | 0.8231 |

**Ablation Study on the Gating Threshold** $\delta$. To assess the sensitivity of the Gating module to the pre-defined error threshold $\delta$, we conduct an ablation on the **DRIVE** dataset by varying $\delta \in \{0.3, 0.4, 0.5, 0.6, 0.7\}$, with results summarized in Table 4. While certain metrics (e.g., SE at

$\delta = 0.3$) are slightly higher, $\delta = 0.5$ delivers the best overall performance across Dice, ACC, AUC, SE, and SP. Intuitively, a too-small threshold treats most pixels as uncertain, triggering unnecessary refinement and reducing gating sparsity, whereas a too-large threshold routes only a few pixels and underutilizes the refinement experts. Balancing these effects, we adopt a fixed threshold of $\delta = 0.5$ across all five public datasets for both accuracy and efficiency.

Table 4: The ablation results of threshold $\delta$ in the Gating module.

| $\delta$ | Dice | ACC | AUC | SE | SP |
|---|---|---|---|---|---|
| 0.3 | 0.8124 | 0.9712 | 0.9812 | **0.8432** | 0.9601 |
| 0.4 | 0.8187 | 0.9755 | 0.9846 | 0.8410 | 0.9732 |
| **0.5** | **0.8231** | **0.9790** | **0.9870** | 0.8366 | **0.9834** |
| 0.6 | 0.8180 | 0.9767 | 0.9854 | 0.8204 | 0.9807 |
| 0.7 | 0.8129 | 0.9735 | 0.9822 | 0.8083 | 0.9784 |

**Effect of number of Residual Experts.** We evaluate the effect of varying the number of Residual Experts ($J = 1, 2, 4, 6$) on the DRIVE dataset, as shown in Tab. 6. Increasing $j$ consistently improves segmentation performance. The Dice score rises from 0.7501 (1 expert) to 0.8231 (6 experts), with corresponding gains across all other metrics. Notably, performance gains begin to saturate beyond $j = 4$, suggesting that using a moderate number of experts balances accuracy and computational efficiency. We adopt $j = 4$ for all remaining experiments as a trade-off between performance and resource consumption.

Table 5: Ablation study of FineSAM++ modules on the DRIVE dataset.

| Lora-SAM | Gating | Residual Experts | Dice | ACC | AUC | SE | SP |
|---|---|---|---|---|---|---|---|
| ✓ | ✗ | ✗ | 0.7322 | 0.9524 | 0.9711 | 0.7865 | 0.9678 |
| ✓ | ✗ | ✓ | 0.7871 | 0.9634 | 0.9821 | 0.8147 | 0.9772 |
| ✓ | ✓ | ✓ | 0.8231 | 0.9790 | 0.9870 | 0.8366 | 0.9834 |

**Qualitative Analysis of Local Structure Refinement.** Fig. 5 presents the FineSAM++ refinement pipeline across multiple vascular segmentation datasets, including fundus, angiography, and OCT-like images. Starting from the coarse LoRA-SAM prediction, which often suffers from topological errors, the Gating Module identifies uncertain regions and selectively routes them to Residual Experts for localized correction. The final output shows improved connectivity and boundary completeness. For visualization clarity, only the first expert's outputs are shown, though FineSAM++ operates with a mixture of experts.

Table 6: Ablation study of the number of Residual Experts ($J$) on the DRIVE dataset.

| Number | Dice | ACC | AUC | SE | SP |
|---|---|---|---|---|---|
| 1 | 0.7501 | 0.956 | 0.9751 | 0.7943 | 0.9723 |
| 2 | 0.7769 | 0.9621 | 0.9803 | 0.8081 | 0.9763 |
| 4 | 0.8025 | 0.9655 | 0.9811 | 0.8139 | 0.9783 |
| 6 | 0.8231 | 0.9790 | 0.9870 | 0.8366 | 0.9834 |

**Ablation study of modules.** Tab. 5 presents the effect of incrementally adding FineSAM++ components. Using only the LoRA-SAM backbone yields limited performance (Dice 0.7322). Adding Residual Experts without the Gating Module, where all $J$ experts are uniformly averaged without spatial weighting, improves performance to 0.7871 Dice by introducing localized correction. However, enabling the full pipeline with the Gating Module further increases performance to 0.8231 Dice and leads to consistent improvements across all metrics. The Gating Module provides a soft spatial routing mechanism that dynamically assigns different weights to each expert's output based on local uncertainty, promoting expert specialization and sparse activation. These results validate that targeted soft routing is critical for maximizing expert effectiveness and minimizing unnecessary corrections in confident regions.

## 5  Conclusion

We presented FineSAM++, a structure-aware sparse expert framework for enhancing foundation segmentation models in fine-grained medical image analysis. By introducing a soft Gating Module with

uncertainty-aware spatial routing and deploying multiple Residual Experts with input perturbation diversity, our method achieves localized structural refinement while maintaining global consistency. Extensive experiments on five vascular segmentation benchmarks demonstrate that FineSAM++ consistently outperforms both classical and SAM-adapted baselines across accuracy and topological continuity. Our results validate the effectiveness of sparse expert activation for addressing localized segmentation failures. Future work will explore dynamic expert allocation, adaptive perturbation strategies, and generalization to other medical and natural image dense prediction tasks.

## Acknowledgments

This work was supported by the National Natural Science Foundation of China (Grant No. 82302300). National Natural Science Foundation of China (62376231), Sichuan Science and Technology Program (2024NSFC0658).

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

# A  Dataset Details and Evaluation Metrics

## A.1  Dataset Details

We conduct experiments on five publicly available medical segmentation datasets. The 2D datasets include DRIVE [23], ROSE [24], FIVES [25], DCA1 [26], and CHUAC [27].

**DRIVE** [23]. The DRIVE dataset consists of 40 retinal fundus images ($584 \times 565$ pixels) for vessel segmentation. We follow the official split of 20 training and 20 testing images.

**ROSE** [24]. The ROSE dataset contains 2D retinal optical coherence tomography angiography (OCTA) scans. We use the ROSE-1 (SVC) subset comprising 30 training and 9 testing images ($304 \times 304$ pixels).

**FIVES** [25]. The FIVES dataset (Fundus Image Vessel Segmentation) provides 800 high-resolution color fundus images ($2048 \times 2048$ pixels) with pixel-level vessel annotations. The dataset is split into 600 training and 200 testing images.

**DCA1** [26]. The DCA1 dataset contains 134 coronary angiography images ($300 \times 300$ pixels). We follow the dataset's standard split with 100 training and 34 testing images.

**CHUAC** [27]. The CHUAC dataset consists of 30 coronary angiography images ($189 \times 189$ pixels) with vessel annotations. Following [28], we split the dataset into 20 training and 10 testing images.

**Synapse Dataset.** The dataset [29] contains 30 subjects for training and 20 subjects for testing with abdominal CT scans. It consists of 13 organs, including 8 organs of Synapse, along with esophagus, inferior vena cava, portal and splenic veins, right and left adrenal gland. Consistent with the partitioning strategy outlined in [30].

## A.2  Compared Methods

We compare FineSAM++ against a comprehensive set of state-of-the-art (SOTA) methods previously reported on the evaluated datasets. The competing approaches are categorized into three groups: CNN-based segmentation models, Transformer-based segmentation models, and foundation model variants.

**(1) CNN-based methods.** Classical and recent CNN architectures include U-Net [31], Attention U-Net [32], U-Net++ [33], CS-Net [34], VSSC Net [35], FR-UNet [36], ResU-Net [37], MedUNAS [38], DUNet [39], HRNet [40], R2U-Net [41], GCN [42], Deeplab V3+ [43], CBAM [44], PSPNet [45], ENet [46], SK-Net [47], SegNet [48], COSFIRE [49], CE-Net [50], OCTA-Net [24], COOF [51], MCDAU-Net [52], and MRC-Net [53].

**(2) Transformer-based methods.** Recent hybrid or fully Transformer architectures include Swin-Unet [54], TransUNet [30], SGAT-Net [55], Retina-TransNet [56], CAViT [57], and G2ViT [58].

**(3) SAM foundation model variants.** To benchmark against foundation model-based baselines, we include SAM Adapter [59], H-SAM [8], AutoSAM [60], SAMed [3].

**(4) Topology-aware segmentation.** the topology-aware adaptation by Gupta et al. [61].

# B  Evaluation metrics

To comprehensively assess the model's performance, we introduce the following evaluation metrics: Dice coefficient (Dice), accuracy (ACC), sensitivity (SE), specificity (SP).

**Dice.** DICE score is a popular metric which measures the area/volumetric overlap between the predicted and ground truth discrete masks. It overcomes the class imbalance problem in the pixel-wise accuracy metric by considering only the foreground classes for measuring the overlap. The higher the DICE, the better the segmentation.

**Accuracy (ACC).** ACC measures the overall correctness of the segmentation results, calculating the proportion of correctly classified pixels or voxels to the total number of pixels or voxels.

**Sensitivity (SE).** SE also known as true positive rate or recall, quantifies the model's ability to correctly identify positive instances, indicating the proportion of true positives correctly classified among all actual positives.

**Specificity (SP).** SP measures the model's ability to correctly identify negative instances, representing the proportion of true negatives correctly classified among all actual negatives.

**Connectivity (C).** Connectivity evaluates the structural consistency between the predicted segmentation and the ground truth. It measures the extent to which the connectivity of predicted regions matches that of the ground truth, ensuring the preservation of continuous structures, particularly in medical images.

**Overlapping Area (A).** Overlapping Area measures the absolute area of intersection between the predicted segmentation and the ground truth. Unlike IOU, it focuses solely on the shared region size, often serving as a supplementary metric for segmentation overlap evaluation.

**Consistency of Vessel Length (L).** L quantifies the similarity in vessel lengths between the predicted segmentation and the ground truth. This metric is particularly critical in vascular structure segmentation, ensuring that the predicted vessels maintain accurate geometric proportions.

**clDice.** A topology-based metric is particularly sensitive to a model's performance on thin structures. This metric evaluates the overlap between predicted and ground truth masks while incorporating the topological features of the segmentation output.

## C  Experiments Results

### C.1  Result on Vessel Segmentation

In this section, we add quantitative comparison on FIVES datasets. As shown in Tables 7, it can be seen that our method has relatively higher evaluation indicators.

Table 7: Quantitative comparison on FIVES datasets. The best results are bolded while the second best are underlined.

| Method | Metric | | | | |
|---|---|---|---|---|---|
| | Dice | ACC | AUC | SP | IOU |
| U-Net | 0.8887 | 0.9866 | 0.9300 | 0.9910 | 0.8077 |
| R2U-Net | 0.8492 | 0.9809 | 0.9238 | 0.9899 | 0.7465 |
| Att Unet | 0.8881 | 0.9868 | 0.9272 | 0.9907 | 0.8073 |
| GCN | 0.9002 | 0.9879 | 0.9399 | 0.9922 | 0.8260 |
| Deeplab V3+ | 0.8856 | 0.9850 | 0.9485 | 0.9933 | 0.8075 |
| SK | 0.8835 | 0.9858 | 0.9334 | 0.9912 | 0.7994 |
| CBAM | 0.8850 | 0.9867 | 0.9226 | 0.9901 | 0.8029 |
| PSPNet | 0.8988 | 0.9878 | 0.9396 | 0.9920 | 0.8235 |
| ENet | 0.8909 | 0.9867 | 0.9409 | 0.9922 | 0.8110 |
| SegNet | 0.8509 | 0.9813 | 0.9244 | 0.9899 | 0.7498 |
| Swin-Unet | 0.9013 | 0.9882 | 0.9402 | 0.9922 | **0.8276** |
| TransU-Net | 0.9037 | 0.9883 | 0.9447 | 0.9928 | 0.8317 |
| SGAT-Net | 0.9051 | 0.9886 | 0.9467 | 0.9933 | 0.8347 |
| SAM Aapter | 0.6313 | 0.8578 | 0.8451 | 0.9081 | 0.4630 |
| H-SAM | 0.6696 | 0.9603 | 0.7851 | 0.9887 | 0.5077 |
| AutoSAM | 0.8817 | 0.9875 | 0.9843 | 0.9921 | 0.7979 |
| SAMed | 0.6750 | 0.9590 | 0.9720 | 0.9840 | 0.5140 |
| **Ours** | **0.9141** | **0.9963** | **0.9961** | **0.9939** | 0.8258 |

### C.2  Generalization beyond vessel segmentation

To validate that our framework generalizes beyond vessel segmentation, we evaluate multi-class abdominal organ segmentation on the Synapse Multi-Organ CT dataset (eight organs). Following prior work [30, 3, 8], we adopt the standard split of 18 training and 12 test volumes and apply the corresponding preprocessing and augmentation protocols. As summarized in Table 8, our method attains the *highest* mean Dice (87.97%) and the *lowest* Hausdorff Distance (HD; 7.89) among all compared methods, surpassing strong baselines such as H-SAM and nnU-Net. Notably, our approach maintains high accuracy on challenging small structures (e.g., pancreas), indicating that **FineSAM++**

preserves strong segmentation quality and robustness when extended to more delicate anatomical targets.

Table 8: Comparison with state-of-the-art models on the Synapse multi-organ CT dataset.

| Method | Spleen | Right Kidney | Left Kidney | Gallbladder | Liver | Stomach | Aorta | Pancreas | Mean Dice (%) | HD |
|--------|--------|--------------|-------------|-------------|-------|---------|-------|----------|---------------|-----|
| TransUNet | 87.23 | 63.13 | 81.87 | 77.02 | 94.08 | 55.86 | 85.08 | 75.62 | 77.48 | 31.69 |
| SwinUNet | 85.47 | 66.53 | 83.28 | 79.61 | 94.29 | 56.58 | 90.66 | 76.60 | 79.13 | 21.55 |
| TransDeepLab | 86.04 | 69.16 | 84.08 | 79.88 | 93.53 | 61.19 | 89.00 | 78.40 | 80.16 | 21.25 |
| DAE-Former | 88.96 | 72.30 | 86.08 | 80.88 | 94.98 | 65.12 | 91.94 | 79.19 | 82.43 | 17.46 |
| MERIT | 92.01 | 84.85 | 87.79 | 74.40 | 95.26 | 85.38 | 87.71 | 71.81 | 84.90 | 13.22 |
| nnU-Net | 91.68 | 88.46 | 83.68 | 70.82 | 97.13 | 83.34 | 93.04 | 81.50 | 87.33 | 10.78 |
| AutoSAM | 80.54 | 80.02 | 79.60 | 41.37 | 89.24 | 61.14 | 82.56 | 44.22 | 62.08 | 27.56 |
| SAM Adapter | 83.68 | 79.00 | 79.02 | 57.49 | 92.67 | 69.48 | 77.93 | 43.07 | 72.80 | 33.08 |
| SAMed | 87.77 | 69.11 | 80.45 | 79.95 | 94.80 | 72.17 | 88.72 | 82.06 | 81.88 | 20.64 |
| H-SAM | 93.34 | 89.93 | 91.88 | 73.49 | 95.72 | 87.10 | 89.38 | 71.11 | 86.49 | 8.18 |
| **Ours** | **94.25** | **91.53** | **93.21** | **71.23** | **96.89** | **90.83** | **92.52** | **82.23** | **87.97** | **7.89** |

# D  Ablation study

## D.1  Effect of Progressive Optimization with Dynamic Weighting

We evaluate the impact of progressive optimization with dynamic uncertainty-based loss weighting by comparing three training strategies: (i) *Naïve Joint Training*, i.e., end-to-end optimization with uniform loss weights; (ii) *Stage-wise (Independent)*, which freezes the coarse module and trains the refinement module separately; and (iii) **Ours (Progressive)**, which progressively optimizes the two modules with dynamic, uncertainty-aware weighting. As summarized in Table 9, the progressive strategy consistently achieves the best overall performance across Dice, ACC, AUC, SE, and SP. We attribute these gains to tighter interaction between the coarse and refinement modules and the reweighting of supervision toward uncertain regions, which improves refinement quality without overfitting.

Table 9: Ablation study on training strategies.

| Strategy | Dice | ACC | AUC | SE | SP |
|----------|------|-----|-----|-----|-----|
| Naïve Joint Training | 0.7984 | 0.9641 | 0.9745 | 0.8123 | 0.9632 |
| Stage-wise (Independent) | 0.8117 | 0.9722 | 0.9810 | 0.8289 | 0.9766 |
| **Ours (Progressive)** | **0.8231** | **0.9790** | **0.9870** | **0.8366** | **0.9834** |

## D.2  Degrade Strategy for Robust Multi-Expert Refinement

Refinement modules may overfit to thin structures, which can increase false positives or degrade mask quality in regions with weak boundaries or ambiguous textures. To mitigate this, we introduce a *Degrade* strategy in **FineSAM++**: rather than feeding all experts the same coarse mask, we apply randomized degradations to the coarse mask (see Eq. (5)) to encourage input diversity and expert specialization. This promotes complementary expertise across residual experts and improves robustness to structural uncertainty. As summarized in Table 10, enabling the Degrade strategy yields consistent gains across Dice, ACC, AUC, and SP, while maintaining competitive SE, indicating fewer false positives and stronger generalization.

Table 10: Ablation study of the *Degrade* strategy for multi-expert training.

| Strategy | Dice | ACC | AUC | SE | SP |
|----------|------|-----|-----|-----|-----|
| No Degrade | 0.8154 | 0.9735 | 0.9813 | 0.8281 | 0.9720 |
| Degrade | **0.8231** | **0.9790** | **0.9870** | 0.8366 | **0.9834** |

# E  Statistical significance.

We assess the statistical significance of our improvements using paired $t$-tests between our method and each baseline across all test images and datasets. Table 11 reports the resulting $p$-values for five

metrics (Dice, ACC, AUC, SE, SP); values in **bold** indicate $p < 0.05$. As shown, the majority of comparisons reach statistical significance, supporting that our approach yields superior segmentation accuracy with improved consistency (lower variance) across datasets.

Table 11: Paired $t$-test $p$-values comparing our method against baselines. Bold indicates statistical significance ($p < 0.05$).

| Dataset | Method | Dice | ACC | AUC | SE | SP |
|---------|--------|------|-----|-----|-----|-----|
| DRIVE | HQ-SAM | **9.86E-03** | **3.99E-04** | **2.47E-12** | 1.55E-01 | 1.32E-01 |
|       | nnU-Net | **5.50E-02** | **9.79E-06** | **2.44E-12** | **3.62E-03** | **2.08E-02** |
|       | RETFound | **1.38E-02** | **9.61E-07** | **1.38E-13** | **5.60E-03** | 5.96E-01 |
| DCAI | HQ-SAM | **4.53E-07** | 9.97E-01 | **1.07E-21** | **3.28E-06** | 2.57E-01 |
|      | nnU-Net | **8.79E-04** | **1.35E-14** | **2.10E-02** | **6.22E-03** | **1.14E-02** |
|      | RETFound | **1.24E-05** | **1.80E-05** | 4.79E-01 | **1.93E-06** | **1.70E-02** |
| CHUAC | HQ-SAM | 1.04E-01 | **8.28E-08** | **1.71E-04** | **9.80E-03** | **6.41E-05** |
|       | nnU-Net | 6.96E-01 | 2.50E-01 | **2.50E-03** | 2.77E-01 | 4.46E-01 |
|       | RETFound | **3.23E-01** | **7.69E-04** | 2.23E-01 | 3.12E-01 | 4.02E-01 |
| ROSE | HQ-SAM | **1.97E-03** | 3.20E-01 | 1.25E-01 | **1.43E-06** | 7.49E-01 |
|      | nnU-Net | 7.03E-01 | 3.13E-01 | 1.65E-01 | **3.02E-04** | 8.90E-01 |
|      | RETFound | **1.64E-04** | **1.51E-03** | 3.11E-01 | **6.43E-06** | **4.93E-02** |

# F   Societal impact discussion

*FineSAM++* is designed to mitigate localized failures in medical image segmentation—particularly in fine-grained regions (e.g., vessels with blurred boundaries)—via a structure-aware, sparsely activated expert mechanism that enhances fidelity with minimal computational overhead. This design advances two practical goals: (i) improving the reliability of automated tools that support clinical decision-making, thereby reducing the risk of missed or spurious findings in delicate anatomical structures; and (ii) enabling a scalable, resource-efficient adaptation strategy that lowers deployment barriers in low-resource healthcare settings where retraining or maintaining large models is impractical.

# G   Limitations

While FineSAM++ demonstrates strong performance and robustness across multiple vascular segmentation benchmarks, several limitations remain. First, the current design employs a fixed number of Residual Experts with pre-defined perturbation settings, which may not fully capture the variability of structural errors across highly diverse anatomical regions. Future work could explore dynamic expert allocation or adaptive degradation strategies conditioned on image content. Second, FineSAM++ introduces additional complexity compared to single-expert or fully fine-tuned models. Third, our study focuses primarily on vascular datasets. Extension to other fine-grained medical segmentation tasks, such as tumor boundary refinement or organ delineation, remains to be validated. We believe these directions provide promising opportunities for further improving the generality and applicability of sparse expert refinement frameworks.

