# OpenReview forum: "Gate to the Vessel: Residual Experts Restore What SAM Overlooks"
_NeurIPS.cc/2025/Conference — NeurIPS 2025 poster_

### Official Review · Reviewer_ospL · 2025-06-30

**Clarity:** 3
**Significance:** 2
**Originality:** 2
**Rating:** 4
**Confidence:** 4

**Summary:**

Segmentation foundation models like SAM demonstrate strong generalization on natural images but tend to underperform on medical images, which often involve fine-grained structures such as blood vessels. This paper introduces FineSAM++, a sparse expert framework designed to refine SAM's output. Specifically, it employs a confidence-driven soft routing module to identify uncertain regions and selectively activate mixed-of-residual-experts to correct structural errors.

**Questions:**

See weakness.

**Ethical Concerns:**

["NO or VERY MINOR ethics concerns only"]

**Final Justification:**

The response from the author solves most of my concerns. I will maintain my initial recommendation on accepting this paper

**Limitations:**

Yes

**Quality:**

3

**Strengths And Weaknesses:**

Strengths:

The proposed framework is well-motivated. While SAM is powerful, it struggles to capture fine-grained, high-resolution details—especially in medical images. Instead of retraining the entire model, this work proposes a more efficient approach: obtain an initial coarse mask, identify uncertain regions, and apply residual corrections through specialized experts. These diverse Residual Experts are trained to address different types of errors, such as disconnections, missing thin vessels, or noisy boundaries.

The paper includes thorough ablation studies that demonstrate:

- the effectiveness of using multiple specialized Residual Experts over a single expert, and

- the benefit of the proposed gating module for selective refinement.

A comprehensive evaluation is conducted across multiple datasets with a wide range of baselines, showing that the proposed method delivers competitive performance.

Weaknesses:

While it is intuitive to perform refinement on top of coarse masks from a foundation model, it would be important to acknowledge and analyze potential failure cases. In my own experience, refinement modules can sometimes overfit to thin structures, leading to false positives and degrading the initial mask quality. Did the authors observe any such failure modes where refinement worsened the output?

The paper lacks an ablation on the progressive optimization strategy with dynamic weighting. It would be helpful to compare this approach with a simpler stage-wise training setup, where each correction module is trained independently. A discussion on the specific advantages of the proposed strategy is needed.

The Residual Experts appear to be relatively independent from the base model. Have the authors explored lightweight, orthogonal integration methods such as adapters? This could offer insights into whether the refinement process is generalizable.

---

> ### Author Rebuttal · Authors · 2025-07-30
>
> We thank reviewer for his/her valuable and insightful reviews. Here we address his/her main concerns:
>
> **W1 “While it is intuitive to perform refinement on top of coarse masks from a foundation model, it would be important to acknowledge and analyze potential failure cases. In my own experience, refinement modules can sometimes overfit to thin structures, leading to false positives and degrading the initial mask quality. Did the authors observe any such failure modes where refinement worsened the output?”:**
>
> We thank the reviewer for the insightful comment. We agree that refinement modules can sometimes overfit to thin structures, potentially leading to false positives or degraded mask quality—especially in regions with weak boundaries or ambiguous textures. This is an important and realistic concern that we have also observed in our experiments.
>
> To address this, we adopt a Degrade strategy in FineSAM++. Instead of feeding all experts the same coarse mask, we introduce randomized degradations:
>
> $$
> \hat{y}_{\text{SAM}}^{(j)}=\text{Degrade} ( {\hat y}\_{SAM}, \eta\_j )
> $$
>
> where $\text{Degrade}(\cdot)$ applies random masking, noise injection, or occlusion perturbations to promote coarse mask input diversity. This encourages specialization across experts and improves robustness to structural uncertainty. We conducted an ablation experiment to evaluate the effectiveness of the degradation mechanism. The results are summarized in Table XV.
>
> **Table XV.** Ablation study of the Degrade strategy for multi-expert training.
> | Strategy      | Dice    | ACC    | AUC    | SE     | SP     |
> |---------------|---------|--------|--------|--------|--------|
> | No Degrade  | 0.8154  | 0.9735 | 0.9813 | 0.8281 | 0.9720 |
> | Degrade    | **0.8231** | **0.9790** | **0.9870** | 0.8366 | **0.9834** |
>
> The Degrade strategy consistently improved robustness, reduced false positives, and encouraged specialization across experts. We will include these results and visual examples in the revised submission and supplemental material.
>
> **W2: The ablation of progressive optimization strategy**
>
> Thank you for the insightful suggestion. We agree that analyzing the effect of the progressive optimization strategy with dynamic weighting is important. To this end, we compared three strategies:
>
> 1）Naïve Joint Training: end-to-end optimization with uniform loss weights.
>
> 2）Stage-wise (Independent): freeze coarse module, train refinement separately.
>
> 3）Ours (Progressive): progressive training with dynamic uncertainty-based loss weighting.
>
> The results are summarized in Table XVI. Our method consistently outperforms both baselines. The progressive setup encourages interaction between modules and focuses learning on uncertain regions, which improves refinement without overfitting. Our method consistently outperforms both baselines. The progressive setup encourages interaction between modules and focuses learning on uncertain regions, which improves refinement without overfitting. We will include these results and discussion in the revised paper. Thank you again for the helpful suggestion. We will include these results and discussion in the revised paper. Thank you again for the helpful suggestion.
>
> **Table. XVI**: Ablation study on training strategies.
> | Strategy                | Dice   | ACC    | AUC    | SE     | SP     |
> |-------------------------|--------|--------|--------|--------|--------|
> | Naïve Joint Training    | 0.7984 | 0.9641 | 0.9745 | 0.8123 | 0.9632 |
> | Stage-wise (Independent)| 0.8117 | 0.9722 | 0.9810 | 0.8289 | 0.9766 |
> | **Ours (Progressive)**  | **0.8231** | **0.9790** | **0.9870** | **0.8366** | **0.9834** |
>
> **W3 “The Residual Experts appear to be relatively independent from the base model. Have the authors explored lightweight, orthogonal integration methods such as adapters? This could offer insights into whether the refinement process is generalizable”:**
>
> We thank the reviewer for the insightful comment. We have indeed explored adapter-based refinement as a baseline, inspired by recent methods such as SAM-Adapter [1]. While adapters provide a parameter-efficient integration mechanism, we found them suboptimal for our task: they struggle to correct fine-grained structures such as thin vessels, bifurcations, and topological continuity.
>
> As shown in Tab. L and Fig. 4 (in manuscript), adapter-based methods often fail to recover fragmented branches or subtle boundaries, especially in high-precision vessel segmentation. These limitations motivated us to explore a mechanism that activates refinement paths only when needed—allowing the model to focus capacity on challenging regions while maintaining overall efficiency.
>
> **Table. X**: Comparison with SAM-Adapter across vessel datasets.
>
> | Dataset | Methods      | **Dice** | **ACC**  | **AUC**  | **SE**   | **SP**   | **C**    | **A**    | **L**    | **ClDice** |
> |---------|--------------|----------|----------|----------|----------|----------|----------|----------|----------|------------|
> | Drive   | SAM-Adapter  | 0.4498   | 0.9311   | 0.9204   | 0.7577   | 0.9377   | 0.993    | 0.412    | 0.321    | 0.488      |
> |         | **Ours**     | 0.8231   | 0.9790   | 0.9870   | 0.8366   | 0.9834   | 0.998    | 0.848    | 0.865    | 0.832      |
> | DCA1    | SAM-Adapter  | 0.7583   | 0.9727   | 0.9408   | 0.7882   | 0.9836   | 0.998    | 0.785    | 0.812    | 0.800      |
> |         | **Ours**     | 0.8127   | 0.9775   | 0.9931   | 0.8479   | 0.9872   | 0.997    | 0.903    | 0.877    | 0.865      |
> | CHUAC   | SAM-Adapter  | 0.7636   | 0.9784   | 0.9359   | 0.7583   | 0.9902   | 0.998    | 0.759    | 0.768    | 0.750      |
> |         | **Ours**     | 0.7768   | 0.9807   | 0.9951   | 0.7567   | 0.9932   | 0.998    | 0.787    | 0.795    | 0.770      |
> | ROSE    | SAM-Adapter  | 0.6316   | 0.8578   | 0.8451   | 0.6503   | 0.9801   | 0.992    | 0.683    | 0.657    | 0.660      |
> |         | **Ours**     | 0.8220   | 0.9483   | 0.9827   | 0.9485   | 0.9823   | 0.997    | 0.819    | 0.839    | 0.805      |
>
> Mixture-of-Experts (MoE) architectures are well-suited for modeling spatially sparse and heterogeneous error patterns [2–4], which are characteristic of SAM’s failure cases in medical imaging. Our confidence-guided routing module enables spatially adaptive specialization by activating lightweight Residual Experts only in uncertain regions. Unlike bottlenecked adapter layers, these CNN-based experts retain local structural priors critical for topology-aware refinement.
>
> Moreover, while SAM-Adapter [1] focuses on global adaptation in natural scenes (e.g., camouflage, shadows), our method is designed for localized, structure-aware correction—essential in clinical settings where boundary precision and topological integrity are crucial. FineSAM++ consistently outperforms both adapter-based and task-specific baselines across five vessel benchmarks, demonstrating the effectiveness and generalizability of this sparse, modular correction paradigm.
>
> References
>
> [1] Chen T., et al (2023). Sam-adapter: Adapting segment anything in underperformed scenes.
>
> [2] Jacobs R A., et al (1991). Adaptive mixtures of local experts[J]. Neural computation.
>
> [3] Fedus W., et al (2022). Switch transformers: Scaling to trillion parameter models with simple and efficient sparsity.
>
> [4] Hwang C., et al (2023). Tutel: Adaptive mixture-of-experts at scale.

---

> > ### Comment · Reviewer_ospL · 2025-08-06
> >
> > Thanks for the detailed response, which solves most of my concerns. I will maintain my initial recommendation on accepting this paper

---

> > > ### Author Response · Authors · 2025-08-07
> > > **Response to Positive Feedback from Reviewer ospL**
> > >
> > > Thank you for your follow-up and for taking the time to carefully consider our responses. We truly appreciate your thoughtful engagement with our work. We're glad to hear that we were able to address most of your concerns, and we sincerely thank you for maintaining your initial recommendation to accept the paper.
> > >
> > > Please don’t hesitate to let us know if there are any remaining points you'd like us to further clarify. We are committed to making the final version of the paper as clear and rigorous as possible.

---

### Official Review · Reviewer_gHFg · 2025-06-30

**Clarity:** 2
**Significance:** 2
**Originality:** 2
**Rating:** 4
**Confidence:** 3

**Summary:**

This paper introduces FineSAM++, a structure-aware sparse expert framework designed to enhance the performance of foundation segmentation models like SAM on medical images, particularly for fine-grained structures such as vessels. FineSAM++ employs a confidence-driven soft Routing Module to dynamically identify uncertain regions and selectively activate a lightweight Residual Expert for local refinement, enabling efficient correction without full retraining.

**Questions:**

See above

**Ethical Concerns:**

["NO or VERY MINOR ethics concerns only"]

**Final Justification:**

As the authors have addressed some of the previous concerns, I have increased my score for acceptance.

**Limitations:**

It is recommended to provide a discussion of the societal impact.

**Quality:**

2

**Strengths And Weaknesses:**

Strengths:
1. The idea of proposed method is simple and straightforward.
2. The proposed method achieves state-of-the-art performance on five publicly available vascular segmentation datasets.

Weaknesses:
1. While SAM demonstrates strong generalization across various diseases and imaging modalities, it would be helpful to clarify why the proposed MoE framework is applied only to vessel segmentation. Extending the approach to other structures or modalities could further validate its generalizability.
2. It is recommended to provide a comparison of the number of trainable parameters and the inference FLOPs between FineSAM++ and baseline methods, to better illustrate the computational efficiency of the proposed framework.
3. The paper would benefit from additional ablation studies or analysis on hyperparameters, such as the impact of $\delta$ in Equation 3, to help understand their influence on performance.
4. A more detailed description of the method, including the design of $g_\theta$  in Equation 2 and the architecture of the expert modules, would improve the clarity and reproducibility of the work.
5. It is suggested to evaluate the effectiveness of FineSAM++ on multi-class segmentation and different disease segmentation tasks to further demonstrate its versatility and robustness.

---

> ### Author Rebuttal · Authors · 2025-07-30
>
> We thank reviewer for his/her valuable and insightful reviews. Here we address his/her main concerns:
>
> **W1, 5 “Extending the approach to other structures or modalities to validate its generalizability.”:**
>
> We agree with the concern regarding the importance of validating the generalizability of our proposed framework beyond vessel segmentation. In response, we have extended our evaluation to multi-class segmentation on the Synapse Multi-Organ CT dataset, which includes eight anatomically diverse abdominal organs. Following [1, 8, 9], we split the dataset into 18 training samples and 12 test samples, and perform corresponding preprocessing and data augmentation methods.
>
> We present the quantitative results in Tab.XII. As shown, our method achieves the highest average Dice score (87.97%) and the lowest Hausdorff Distance (HD) of 7.89 among all compared methods, outperforming strong baselines such as H-SAM and nnU-Net. Notably, our method maintains high segmentation accuracy on challenging small organs like the pancreas, demonstrating that FineSAM++ preserves its segmentation strength and robustness even when extended to more delicate anatomical structures. We will include this supplementary analysis in the final submission to better showcase the generalization capability of our approach and further support the robustness of the method.
>
> **Table XI.** Comparison to state-of-the-art models on Synapse multi-organ CT dataset.
>
> | Method        | Spleen | Right Kidney | Left Kidney | Gallbladder | Liver | Stomach | Aorta | Pancreas | Mean Dice (%) | HD    |
> |---------------|--------|--------------|-------------|-------------|-------|---------|-------|----------|----------------|--------|
> | TransUnet [1]     | 87.23  | 63.13        | 81.87       | 77.02       | 94.08 | 55.86   | 85.08 | 75.62    | 77.48          | 31.69  |
> | SwinUnet [2]     | 85.47  | 66.53        | 83.28       | 79.61       | 94.29 | 56.58   | 90.66 | 76.6     | 79.13          | 21.55  |
> | TransDeepLab [3] | 86.04  | 69.16        | 84.08       | 79.88       | 93.53 | 61.19   | 89.00 | 78.4     | 80.16          | 21.25  |
> | DAE-Former [4]   | 88.96  | 72.3         | 86.08       | 80.88       | 94.98 | 65.12   | 91.94 | 79.19    | 82.43          | 17.46  |
> | MERIT [5]        | 92.01  | 84.85        | 87.79       | 74.4        | 95.26 | 85.38   | 87.71 | 71.81    | 84.9           | 13.22  |
> | nnUnet [10]       | 91.68  | 88.46        | 83.68       | 70.82       | 97.13 | 83.34   | 93.04 | 81.5     | 87.33          | 10.78  |
> | AutoSAM [6]      | 80.54  | 80.02        | 79.6        | 41.37       | 89.24 | 61.14   | 82.56 | 44.22    | 62.08          | 27.56  |
> | SAM Adapter [7]  | 83.68  | 79.00        | 79.02       | 57.49       | 92.67 | 69.48   | 77.93 | 43.07    | 72.8           | 33.08  |
> | SAMed [8]        | 87.77  | 69.11        | 80.45       | 79.95       | 94.8  | 72.17   | 88.72 | 82.06    | 81.88          | 20.64  |
> | H-SAM [9]        | 93.34  | 89.93        | 91.88       | 73.49       | 95.72 | 87.1    | 89.38 | 71.11    | 86.49          | 8.18   |
> | **Ours**      | **94.25** | **91.53**    | **93.21**   | **71.23**   | **96.89** | **90.83** | **92.52** | **82.23**  | **87.97**        | **7.89** |
>
> **W2 “Parameter efficiency”:**
>
> Thanks for your valuable comment. To address your concern regarding the efficiency of our proposed method FineSAM++ in terms of learnable parameters among SAM-based segmentation methods, we have conducted a detailed comparison. Specifically, we computed the number of parameters, FLOPs (GAMCs), and inference latency under a standard input resolution of (1, 3, 1024, 1024). The results are summarized in the Tab.X below:
>
> **Table X.** Comparison of model size, computational cost, latency, and segmentation accuracy (Dice score) across segmentation methods using an input of size (1, 3, 1024, 1024).
>
> | **Model**     | **Params (M)**                | **GAMCs (G)** | **Latency (ms)** | **Dice**  |
> |---------------|-------------------------------|----------------|------------------|-----------|
> | Unets         | 34.53                         | 4.08           | 1.10             | 0.7787    |
> | nnUnet        | 126.2                         | 1864.9         | 37.4             | 0.8220    |
> | SAM Adapter   | 104.3                         | 400.1          | 127.8            | 0.4498    |
> | H-SAM         | 111.3                         | 370.6          | 124.8            | 0.6622    |
> | AutoSAM       | 135.29                        | 774.16         | 166.22           | 0.6603    |
> | SAMed         | 92.2                          | 370.5          | 117.1            | 0.6170    |
> | SAM           | 93.7                          | 372.0          | 116.33           | /         |
> | Ours (FineSAM++) | 94.4 | 376.8 | 117.6| 0.8231 |
>
> As shown in the Tab.X, FineSAM++ introduces only 1.4M additional learnable parameters on top of the SAM backbone. Compared to other SAM-based segmentation methods, FineSAM++ demonstrates significantly higher parameter efficiency. Although the total parameter count is higher than lightweight models like Unet, FineSAM++ achieves the highest Dice score (0.8231). These results clearly indicate that FineSAM++ strikes an excellent balance between parameter efficiency and segmentation performance. We will include this comparison in the revised version.
>
> **W3 “Ablation study on threshold** $\delta$ **in the Gating module”:**
>
> Thanks for pointing this out. To assess the sensitivity of the Gating module to the pre-defined error threshold $\delta$, we conducted an ablation study on the DRIVE dataset by varying $\delta \in \{0.3, 0.4, 0.5, 0.6, 0.7\}$. The results are summarized in the Tab.XI below:
>
> **Table XI**. The Ablation Result of threshold δ in the Gating module
> | **δ**      | **Dice** | **ACC** | **AUC** | **SE**        | **SP**        |
> |------------|----------|---------|---------|---------------|---------------|
> | 0.3        | 0.8124   | 0.9712  | 0.9812  | **0.8432**    | 0.9601        |
> | 0.4        | 0.8187   | 0.9755  | 0.9846  | 0.8410        | 0.9732        |
> | **0.5**    | **0.8231** | **0.9790** | **0.9870** | 0.8366        | **0.9834**    |
> | 0.6        | 0.8180   | 0.9767  | 0.9854  | 0.8204        | 0.9807        |
> | 0.7        | 0.8129   | 0.9735  | 0.9822  | 0.8083        | 0.9784        |
>
> As shown, while some metrics are slightly higher, $\delta=0.5$ achieves the best overall performance across all metrics. Overall, $\delta=0.5$ provides a meaningful routing threshold:
>
> - If $\delta$ is too low, almost all pixels are considered uncertain, resulting in unnecessary refinement and loss of gating sparsity.
> - If $\delta$ is too high, only a few pixels are routed, making the refinement module underutilized.
>
> Thus, $\delta=0.5$ strikes a balance, activating residual experts in genuinely ambiguous regions while maintaining efficiency. We will include this explanation and the ablation results in the final version.
>
> **W4 “A more detailed description of the method, including the design of $g_\theta$ in Equation 2 and the architecture of the expert modules”:**
>
> We thank the reviewer for emphasizing the importance of methodological clarity. We will revise the manuscript to provide a more detailed description.
>
> **1.** Gating Module $g_\theta$ (Eq. 2):
>
> $g_\theta$ is implemented as a lightweight CNN without downsampling. It contains three convolutional layers with kernel size $3 \times 3$, followed by BatchNorm and ReLU activation. It produces two outputs through separate heads:
>
> •	Uncertainty map: A $1 \times 1$ convolution followed by a sigmoid activation outputs a spatial confidence map.
>
> •	Routing weights: The feature map is globally pooled and passed through an MLP to produce per-image expert routing weights.
>
> **2.** Expert Modules:
>
> Each expert is a compact 3-level U-Net, where each level contains 2 down-sampling and 2 up-sampling blocks. All experts share architecture but are independently trained to specialize in different error patterns. The final prediction is a weighted sum of expert outputs, gated by the spatial uncertainty and routing weights. We will update the final paper and appendix to support reproducibility.
>
> **L1 “Societal impact discussion”:**
>
> We appreciate the reviewer’s suggestion to address societal impact. FineSAM++ targets localized failures in medical images, especially in fine-grained regions like vessels with blurred boundaries, by introducing a structure-aware, sparsely activated expert mechanism that improves segmentation fidelity with minimal overhead. This design offers two key benefits: (1) improved reliability of automated tools to support clinical decision-making, and (2) a scalable, resource-efficient adaptation strategy that facilitates deployment in low-resource healthcare settings where retraining large models is impractical.
>
> References
>
> [1] Chen J., et al (2021). Transunet: Transformers make strong encoders for medical image segmentation.
>
> [2] Cao H., et al (2022). Swin-unet: Unet-like pure transformer for medical image segmentation.
>
> [3] Azad R., et al (2022). Transdeeplab: Convolution-free transformer-based deeplab v3+ for medical image segmentation.
>
> [4] Azad R., et al (2023). Dae-former: Dual attention-guided efficient transformer for medical image segmentation.
>
> [5] Rahman M M., et al (2024). Multi-scale hierarchical vision transformer with cascaded attention decoding for medical image segmentation.
>
> [6] Hu X., et al (2023). How to efficiently adapt large segmentation model (sam) to medical images.
>
> [7] Chen T., et al (2023). Sam fails to segment anything?–sam-adapter: Adapting sam in underperformed scenes: Camouflage, shadow, and more.
>
> [8] Zhang K., et al (2023). Customized segment anything model for medical image segmentation.
>
> [9] Cheng Z., et al (2024). Unleashing the potential of sam for medical adaptation via hierarchical decoding.
>
> [10] Isensee F., et al (2021). nnU-Net: a self-configuring method for deep learning-based biomedical image segmentation.

---

> > ### Comment · Reviewer_gHFg · 2025-08-08
> >
> > Thank you for your response. I am pleased to see that my concerns have been addressed. I am willing to increase my score accordingly.

---

> ### Author Response · Authors · 2025-08-08
> **Appreciation for Reviewer’s Reconsideration**
>
> We are grateful for your thoughtful reconsideration and for acknowledging the clarifications we provided. Your constructive comments have been valuable in improving the paper.

---

### Official Review · Reviewer_LmaJ · 2025-07-02

**Clarity:** 4
**Significance:** 2
**Originality:** 2
**Rating:** 4
**Confidence:** 3

**Summary:**

The authors build on SAM to produce more coherent vessel segmentations. They introduce two modules: 1. A Routing module to identify uncertain regions. 2. A Residual Expert model (lightweight) that fixes the structures identified by the routing model.
They evaluate their method on 5 vessel datasets on many vessel/tube specific metrics including CLDice and connectivity.

**Questions:**

- Why did the authors choose different baselines per dataset ? It seems to weaken their evaluation and makes performance less comparable
- What is the number of parameters with their suggested changes and how does it compare to other baselines ? (Especially SAM and UNets)
- Why not mention nnUNets, and models specialized for tubular structures ?
- Were the augmentations applied to all baselines as well?
- Would it be possible to see standard deviations to assess the significance of the results?

**Ethical Concerns:**

["NO or VERY MINOR ethics concerns only"]

**Final Justification:**

I appreciate the authors effort to address all my concerns. The marginal improvements on nnUNet seem a bit concerning (sometimes are insignificance). If the main claim is latency, this should be more emphasized in the paper. I think the improvements in latency are interesting though, but should be clarified in the paper.

**Limitations:**

The authors mentioned the added complexity and computation time of their method in the supplemental section. I think it would be great to quantify this numerically both in terms of parameters and in terms of training time (for both phases).

**Quality:**

2

**Strengths And Weaknesses:**

*Strengths*
- the authors evaluate on various datasets and modalities
- a comprehensive set of metrics is used to assess both accuracy and topology of the produced segmentations.
- the authors compare against a comprehensive set of baselines, including specialized models and UNets
- the authors ran an ablation study to understand the significance of each of their module.

*Weaknesses*.

Sparse related work section.
- I would have expected more work on uncertainty evaluation, specialised models and foundation models.
- No mention of the vessel specific literature.

Confusion in the experiments.

- Different baselines are used for different datasets. Why is this the case? Can the author elaborate? I would guess that this limits the strength of the paper.
- Why are the UNet results so variable depending on the dataset?
- Did you also run data augmentations for the baselines?
It seems to me that some important baselines are missing:
- SAM models specifically designed for fine structures such as _Segment anything in high quality_ [1]
- nnUNet, which are a standard network for medical image segmentation [2]
- Foundation model for tubular structures

[1]: Ke, Lei, et al. "Segment anything in high quality." Advances in Neural Information Processing Systems 36 (2023): 29914-29934.
[2]: Isensee, Fabian, et al. "nnU-Net: a self-configuring method for deep learning-based biomedical image segmentation." Nature methods 18.2 (2021): 203-211.

---

> ### Author Rebuttal · Authors · 2025-07-29
>
> We thank reviewer for his/her valuable and insightful reviews. Here we address his/her main concerns:
>
> **W1 “Sparse related work section”:**
>
> We appreciate the reviewer’s observation regarding the breadth of the related work. We would like to clarify that our Related Work section (Sec. 2) will include dedicated subsections that cover foundation models, uncertainty modeling, specialized approaches, and vessel-specific segmentation literature:
>
> **1)Foundation Models & Uncertainty modeling**: We will expand Section 2.1 to cover additional efforts on adapting foundation models for medical image segmentation, including LoRA-based fine-tuning [1], prompt-driven adaptation [2], and SAM-Adapter methods [3]. To address the reviewer’s interest in uncertainty and boundary quality, we will add HQ-SAM [4], which improves boundary precision via high-quality priors and edge-aware refinement. We will also discuss topology-aware uncertainty modeling [5,6], which enhances consistency by quantifying structural uncertainty. These additions better contextualize the potential and limitations of foundation models in segmenting fine-grained medical structures like vessels.
>
> **2) Specialized Models for Vessel Segmentation**: We will expand Section 2.3 to include recent progress in both traditional and foundation model-based vessel segmentation. Beyond CNN and Transformer methods [7,8], we will add vesselFM [9], a foundation model for universal 3D vessel segmentation, and a retinal disease foundation model [10], which supports generalizable vessel-pathology detection. These works underscore the increasing relevance of foundation models in vascular imaging for both segmentation and clinical insight.
>
> **Q1, 3, 4, 5 & W2.1, 2.3, 2.4, 2.5, 2.6 “Baseline selection, data augmentations, missing import baselines and standard deviations”:**
>
> We thank the reviewer for raising this important point. For each dataset, we selected baselines that are both highly cited on Google Scholar and representative of state-of-the-art performance, aiming to reflect the most influential methods specific to each task. Importantly, we included several SAM-based methods (SAM-Adapter, H-SAM, AutoSAM, SAMed) across all datasets, along with a vessel-specific baseline (Gupta et al. [5]). We agree that a unified suite of baselines would improve consistency, and we will revise our experiments to add some important baselines:
>
> 1)	HQ-SAM [4]: A SAM variant specifically designed for fine structural segmentation.
> 2)	nnUNet [11]: A standard network for medical image segmentation.
> 3)	RETFound (modified) [10]: A foundation model for vessel disease detection. We adapt it for segmentation by replacing the classification head with a lightweight decoder.
>
> Furthermore, to ensure fairness, all methods were trained with the same data augmentation strategies, following [5], which includes random rotation, flipping, elastic deformation, and brightness/contrast jittering.
>
> As shown in Table XII and Table 1, HQ-SAM improves over other SAM-based models (e.g., H-SAM, SAM-Adapter, SAMed) via lightweight token-level refinement. However, its frozen encoder and global adaptation lack domain-specific priors and struggle with fine-grained structures such as thin vessels and bifurcations. RETFound performs better structurally due to pretraining, while nnUNet remains stable due to its self-configuring pipeline and strong inductive bias in medical segmentation. Our method achieves balanced and robust performance across datasets, especially on Dice and AUC, critical for fine vessel structures. A more detailed analysis will be given in the revised version.
>
> To better evaluate the robustness of our method, we report the mean ± standard deviation in Table XII. Minor revisions will be made in the final version to include these statistics for completeness.
>
> **Table XII.** Quantitative comparison on DRIVE, DCAI, CHUAC and ROSE datasets.
> | **Dataset** | **Method**    | **Dice**           | **ACC**            | **AUC**            | **SE**             | **SP**             |
> |---------|-----------|----------------|----------------|----------------|----------------|----------------|
> | DRIVE   | HQ-SAM    | 0.7978         | 0.9697         | 0.8824         | 0.8033         | 0.9824         |
> |         | nnUnet    | 0.822          | 0.9698         | 0.894          | 0.8019         | 0.9862         |
> |         | RETFound  | 0.802          | 0.9649         | 0.883          | 0.7796         | 0.9821         |
> |         | Ours      | **0.8231±0.014** | **0.979±0.003** | **0.987±0.023** | **0.8366±0.048** | **0.9834±0.004** |
> | DCAI    | HQ-SAM    | 0.788          | 0.977          | 0.889          | 0.789          | 0.988          |
> |         | nnUnet    | 0.8045         | 0.9584         | 0.9903         | 0.8264         | 0.9879         |
> |         | RETFound  | 0.7938         | 0.9681         | 0.9911         | 0.8853         | 0.9861         |
> |         | Ours      | **0.8127**     | **0.9775**     | **0.9931**     | **0.8479**     | **0.9872**     |
> | CHUAC   | HQ-SAM    | 0.705          | 0.894          | 0.812          | 0.676          | 0.948          |
> |         | nnUnet    | 0.7814         | 0.9776         | 0.8842         | 0.7788         | 0.9896         |
> |         | RETFound  | 0.7636         | 0.9604         | 0.9904         | 0.7325         | 0.9906         |
> |         | Ours      | **0.7768±0.045** | **0.9807±0.005** | **0.9951±0.032** | **0.7567±0.065** | **0.9932±0.003** |
> | ROSE    | HQ-SAM    | 0.752          | 0.9609         | 0.9904         | 0.794          | 0.9887         |
> |         | nnUnet    | 0.827          | 0.947          | 0.931          | 0.865          | 0.994          |
> |         | RETFound  | 0.7126         | 0.9197         | 0.9337         | 0.8563         | 0.9193         |
> |         | Ours      | **0.822±0.047** | **0.9483±0.027** | **0.9827±0.046** | **0.9485±0.047** | **0.9823±0.094** |
>
> **W2.2 “Why are the UNet results so variable depending on the dataset?”:**
>
> Thank you for the question. The performance variability of UNet reflects both dataset characteristics and inherent model limitations. UNet relies heavily on low-level intensity cues and lacks global context modeling, making it sensitive to modality shifts and low-contrast scenarios. For instance, it performs well on DRIVE (high-contrast RGB) but struggles on CHUAC (grayscale, noisy angiography). This highlights UNet’s limited robustness across diverse medical domains.
>
> **Q2 & L1 “Parameter Count Comparison”:**
>
> We thank the reviewers for their comments. To better illustrate the computational properties of our method, we compare its performance against SAM-based methods, UNets, and nnUNet. While training time is a relevant consideration, it is not the primary bottleneck or design objective of our approach. FineSAM++ is built upon a frozen backbone and introduces two lightweight adaptation stages, both of which converge efficiently in practice. Accordingly, we focus on parameter efficiency, inference cost, and segmentation accuracy. The results are summarized in Table X below:
>
> **Table X.** Comparison of model size, computational cost, latency, and segmentation accuracy (Dice score) across segmentation methods using an input of size (1, 3, 1024, 1024).
>
> | **Model**     | **Params (M)**                | **GAMCs (G)** | **Latency (ms)** | **Dice**  |
> |---------------|-------------------------------|----------------|------------------|-----------|
> | Unets         | 34.53                         | 4.08           | 1.10             | 0.7787    |
> | nnUnet        | 126.2                         | 1864.9         | 37.4             | 0.8220    |
> | SAM Adapter   | 104.3                         | 400.1          | 127.8            | 0.4498    |
> | H-SAM         | 111.3                         | 370.6          | 124.8            | 0.6622    |
> | AutoSAM       | 135.29                        | 774.16         | 166.22           | 0.6603    |
> | SAMed         | 92.2                          | 370.5          | 117.1            | 0.6170    |
> | SAM           | 93.7                          | 372.0          | 116.33           | /         |
> | Ours (FineSAM++) | 94.4 | 376.8 | 117.6| 0.8231 |
>
> As shown above, FineSAM++ achieves a Dice score of 0.8231, the highest among all SAM-based methods, while introducing only 1.4M additional learnable parameters. Compared to UNet (1.1 ms latency) and SAM (116.33 ms latency), our method maintains a comparable inference time (117.6 ms) and significantly improves performance over Unet. In contrast to nnUNet, which requires heavy computation (1864.9G FLOPs), FineSAM++ achieves similar accuracy (0.8231 vs. 0.8220) with over 4× fewer parameters and over 30× faster inference.
>
> References:
>
> [1] Cheng Z., et al (2024). Unleashing the potential of sam for medical adaptation via hierarchical decoding.
>
> [2] Chen Z., et al (2025). UN-SAM: Domain-adaptive self-prompt segmentation for universal nuclei images.
>
> [3] Wu J., et al (2025). Medical sam adapter: Adapting segment anything model for medical image segmentation.
>
> [4] Ke L., et al (2023). Segment anything in high quality.
>
> [5] Gupta S., et al (2023). Topology-aware uncertainty for image segmentation.
>
> [6] Huang J., et al (2024). Representing topological self-similarity using fractal feature maps for accurate segmentation of tubular structures.
>
> [7] Qi Y., et al (2023). Dynamic snake convolution based on topological geometric constraints for tubular structure segmentation.
>
> [8] Mou L., et al (2024). CS-Net: Channel and spatial attention network for curvilinear structure segmentation.
>
> [9] Wittmann B., et al (2025). vesselFM: A Foundation Model for Universal 3D Blood Vessel Segmentation.
>
> [10] Zhou Y., et al (2023). A foundation model for generalizable disease detection from retinal images.
>
> [11] Isensee F., et al (2021). nnU-Net: a self-configuring method for deep learning-based biomedical image segmentation.

---

> > ### Comment · Reviewer_LmaJ · 2025-08-05
> > **Thoughts on Standard Deviation and nnUNets**
> >
> > I appreciate the authors’ in depth response and their effort to address my concerns and add in-depth related work. Do the authors have access to the standard deviation for the baselines ? Also, it looks like the nnUNet and FineSAM++ are very close. Do they have any way to validate the statistical significance of their results ? I am also not sure I understand the claims about faster inference times and number of parameters (4x and 30x). They seem inaccurate? Also how are the Dice score computed and aggregated in Table X?

---

> ### Author Response · Authors · 2025-08-06
> **Standard Deviation Reporting, Statistical Significance Analysis, and Clarification of Efficiency Claims**
>
> We appreciate the reviewer’s feedback and addresss the concerns as follows:
>
> **“1.Standard deviation”:**
>
> We have calculated the standard deviation for baseline methods. This addition allows for more reliable comparisons in terms of both performance and stability (See Tab.XII).
>
> Table XII. Quantitative comparison.
>
> | Dataset | Method   | Dice            | ACC             | AUC             | SE              | SP              |
> |------------|--------------|---------------------|---------------------|---------------------|---------------------|---------------------|
> | DRIVE      | HQ-SAM       | 0.7978 ± 0.027      | 0.9697 ± 0.009      | 0.8824 ± 0.021      | 0.8033 ± 0.030      | 0.9824 ± 0.011      |
> |            | nnUNet       | 0.8220 ± 0.017      | 0.9698 ± 0.006      | 0.8940 ± 0.014      | 0.8019 ± 0.025      | 0.9862 ± 0.008      |
> |            | RETFound     | 0.8020 ± 0.022      | 0.9649 ± 0.008      | 0.8830 ± 0.018      | 0.7796 ± 0.028      | 0.9821 ± 0.010      |
> |            | **Ours**         | 0.8231 ± 0.014  | 0.9790 ± 0.003  | 0.9870 ± 0.023  | 0.8366 ± 0.048  | 0.9834 ± 0.004  |
> | DCAI       | HQ-SAM       | 0.7880 ± 0.026      | 0.9770 ± 0.006      | 0.8890 ± 0.018      | 0.7890 ± 0.025      | 0.9880 ± 0.006      |
> |            | nnUNet       | 0.8045 ± 0.015      | 0.9584 ± 0.005      | 0.9903 ± 0.006      | 0.8264 ± 0.021      | 0.9879 ± 0.004      |
> |            | RETFound     | 0.7938 ± 0.021      | 0.9681 ± 0.007      | 0.9911 ± 0.005      | 0.8853 ± 0.019      | 0.9861 ± 0.007      |
> |            | **Ours**         | 0.8127 ± 0.012  | 0.9775 ± 0.003  | 0.9931 ± 0.004  | 0.8479 ± 0.028  | 0.9872 ± 0.005  |
> | CHUAC      | HQ-SAM       | 0.7050 ± 0.030      | 0.8940 ± 0.015      | 0.8120 ± 0.025      | 0.6760 ± 0.035      | 0.9480 ± 0.010      |
> |            | nnUNet       | 0.7814 ± 0.020      | 0.9776 ± 0.004      | 0.8842 ± 0.017      | 0.7788 ± 0.032      | 0.9896 ± 0.003      |
> |            | RETFound     | 0.7636 ± 0.028      | 0.9604 ± 0.008      | 0.9904 ± 0.006      | 0.7325 ± 0.045      | 0.9906 ± 0.004      |
> |            | **Ours**         | 0.7768 ± 0.045  | 0.9807 ± 0.005  | 0.9951 ± 0.032  | 0.7567 ± 0.065  | 0.9932 ± 0.003  |
> | ROSE       | HQ-SAM       | 0.7520 ± 0.028      | 0.9609 ± 0.006      | 0.9904 ± 0.005      | 0.7940 ± 0.021      | 0.9887 ± 0.006      |
> |            | nnUNet       | 0.8270 ± 0.018      | 0.9470 ± 0.006      | 0.9310 ± 0.012      | 0.8650 ± 0.020      | 0.9940 ± 0.005      |
> |            | RETFound     | 0.7126 ± 0.032      | 0.9197 ± 0.009      | 0.9337 ± 0.009      | 0.8563 ± 0.033      | 0.9193 ± 0.007      |
> |            |**Ours**         | 0.8220 ± 0.047  | 0.9483 ± 0.027  | 0.9827 ± 0.046  | 0.9485 ± 0.047  | 0.9823 ± 0.094  |
>
> **“2.Statistical significance”:**
>
> We performed paired t-tests to assess statistical significance (Tab.XIII), where most improvements are significant (p < 0.05 are bold). Our method demonstrates superior overall performance in segmentation accuracy, consistency (lower standard deviation), and statistical significance.
>
> Table XIII. Statistical significance.
>
> | Dataset | Method     | Dice       | ACC        | AUC        | SE         | SP         |
> |---------|------------|------------|------------|------------|------------|------------|
> | DRIVE   | HQ-SAM     | **9.86E-03** | **3.99E-04** | **2.47E-12** | 1.55E-01   | 1.32E-01   |
> |         | nnUNet     | **5.50E-02** | **9.79E-06** | **2.44E-12** | **3.62E-03** | **2.08E-02** |
> |         | RETFound   | **1.38E-02** | **9.61E-07** | **1.38E-13** | **5.60E-03** | 5.96E-01   |
> | DCAI    | HQ-SAM     | **4.53E-07** | 9.97E-01   | **1.07E-21** | **3.28E-06** | 2.57E-01   |
> |         | nnUNet     | **8.79E-04** | **1.35E-14** | **2.10E-02** | **6.22E-03** | **1.14E-02** |
> |         | RETFound   | **1.24E-05** | **1.80E-05** | 4.79E-01   | **1.93E-06** | **1.70E-02** |
> | CHUAC   | HQ-SAM     | 1.04E-01   | **8.28E-08** | **1.71E-04** | **9.80E-03** | **6.41E-05** |
> |         | nnUNet     | 6.96E-01   | 2.50E-01   | **2.50E-03** | 2.77E-01   | 4.46E-01   |
> |         | RETFound   | **3.23E-01** | **7.69E-04** | 2.23E-01   | 3.12E-01   | 4.02E-01   |
> | ROSE    | HQ-SAM     | **1.97E-03** | 3.20E-01   | 1.25E-01   | **1.43E-06** | 7.49E-01   |
> |         | nnUNet     | 7.03E-01   | 3.13E-01   | 1.65E-01   | **3.02E-04** | 8.90E-01   |
> |         | RETFound   | **1.64E-04** | **1.51E-03** | 3.11E-01   | **6.43E-06** | **4.93E-02** |
>
> **“3.Examination of Tab.X”:**
>
> We address the reviewer’s concerns on Tab.X below.
>
> a.Parameter and Latency:
>
> Compared to nnUNet (1864.9G GAMCs), FineSAM++ is ~4.9× more computationally efficient (376.8G GAMCs). The original “30×” was a typographical error based on preliminary theoretical FLOP ratios across scales. We will revise the manuscript.
>
> b.Dice:
>
> The Dice in Tab.X are averaged over all test images per dataset. While our method has higher latency than nnUNet, it outperforms SAM-based baselines with a similar parameter budget.

---

### Official Review · Reviewer_6m4f · 2025-07-02

**Clarity:** 3
**Significance:** 3
**Originality:** 3
**Rating:** 4
**Confidence:** 4

**Summary:**

This paper presents a structure-aware sparse expert framework, named FineSAM++, which enhances foundation segmentation models. The proposed method has a gating module that identifies structurally uncertain regions and activates a Residual Expert to correct local structural inconsistencies of vessel segmentation. The proposed method has been demonstrated on five vessel segmentation datasets, and the results show superior performance compared to the existing state-of-the-art segmentation methods.

**Questions:**

Please refer to the above weaknesses.

- Is the proposed method efficient in terms of the number of learnable parameters among the SAM-based segmentation methods?
- How is the error threshold (\delta) in the gating module defined? Is the threshold the same for all five public datasets?
- The hyperparameter study would strengthen the contribution of the proposed method.

**Ethical Concerns:**

["NO or VERY MINOR ethics concerns only"]

**Final Justification:**

After reviewing the rebuttal by the authors and other reviewers' comments, I keep my rating. The authors have addressed most of my concerns, and the SAM-based proposed method would be helpful in vessel segmentation masks of various medical images.

**Limitations:**

yes (in supplementary material)

**Paper Formatting Concerns:**

There is no concern on the paper formatting.

**Quality:**

3

**Strengths And Weaknesses:**

### Strengths
- The proposed framework with a gating module and residual expert refines the vessel segmentation mask from SAM model effectively.
- The proposed method trains a part of the neural networks, which enables efficient training of the large model.
- Experimental results using five public datasets demonstrate the effectiveness of the proposed method in local correction of the segmentation mask.

### Weaknesses
- There is no comparison for the size of learnable parameters between the proposed method and the existing SAM-based methods.
- The proposed gating module requires a pre-defined error threshold (\delta), which may make the model sensitive to the threshold value.
- There is no study on the hyperparameters used in the gating module and residual experts, such as normalized routing weights (I, w_j) and the threshold (\delta).

---

> ### Author Rebuttal · Authors · 2025-07-30
>
> We thank reviewer for his/her valuable and insightful reviews. Here we address his/her main concerns:
>
> **Q1&W1 “Parameter efficiency vs. SAM-based method”**:
>
> Thanks for your valuable comment. To address your concern regarding the efficiency of our proposed method FineSAM++ in terms of learnable parameters among SAM-based segmentation methods, we have conducted a detailed comparison. Specifically, we computed the number of parameters, FLOPs (GAMCs), and inference latency under a standard input resolution of (1, 3, 1024, 1024). The results are summarized in the Tab.X below:
>
> **Table X.** Comparison of model size, computational cost, latency, and segmentation accuracy (Dice score) across segmentation methods using an input of size (1, 3, 1024, 1024).
>
> | **Model**     | **Params (M)**                | **GAMCs (G)** | **Latency (ms)** | **Dice**  |
> |---------------|-------------------------------|----------------|------------------|-----------|
> | Unets         | 34.53                         | 4.08           | 1.10             | 0.7787    |
> | nnUnet        | 126.2                         | 1864.9         | 37.4             | 0.8220    |
> | SAM Adapter   | 104.3                         | 400.1          | 127.8            | 0.4498    |
> | H-SAM         | 111.3                         | 370.6          | 124.8            | 0.6622    |
> | AutoSAM       | 135.29                        | 774.16         | 166.22           | 0.6603    |
> | SAMed         | 92.2                          | 370.5          | 117.1            | 0.6170    |
> | SAM           | 93.7                          | 372.0          | 116.33           | /         |
> | Ours (FineSAM++) | 94.4 | 376.8 | 117.6| 0.8231 |
>
> As shown in the Tab.X, FineSAM++ introduces only 1.4M additional learnable parameters on top of the SAM backbone. Compared to other SAM-based segmentation methods—such as SAM Adapter (104.3M), H-SAM (111.3M), and AutoSAM (135.29M)—FineSAM++ demonstrates significantly higher parameter efficiency. Although the total parameter count is higher than lightweight models like Unet, FineSAM++ achieves the highest Dice score (0.8231). These results clearly indicate that FineSAM++ strikes an excellent balance between parameter efficiency and segmentation performance. We will include this comparison in the revised version.
>
> **Q2&W2 “Ablation study on threshold** $\delta$ **in the Gating module”:**
>
> Thank you for pointing this out. To assess the sensitivity of the Gating module to the pre-defined error threshold $\delta$, we conducted an ablation study on the DRIVE dataset by varying $\delta \in \{0.3, 0.4, 0.5, 0.6, 0.7\}$. The results are summarized in the Tab.XI below:
>
> **Table XI**. The Ablation Result of threshold δ in the Gating module
> | **δ**      | **Dice** | **ACC** | **AUC** | **SE**        | **SP**        |
> |------------|----------|---------|---------|---------------|---------------|
> | 0.3        | 0.8124   | 0.9712  | 0.9812  | **0.8432**    | 0.9601        |
> | 0.4        | 0.8187   | 0.9755  | 0.9846  | 0.8410        | 0.9732        |
> | **0.5**    | **0.8231** | **0.9790** | **0.9870** | 0.8366        | **0.9834**    |
> | 0.6        | 0.8180   | 0.9767  | 0.9854  | 0.8204        | 0.9807        |
> | 0.7        | 0.8129   | 0.9735  | 0.9822  | 0.8083        | 0.9784        |
>
> As shown, while some metrics (e.g., SE at $\delta=0.3$) are slightly higher, $\delta=0.5$ achieves the best overall performance across all metrics. Overall, $\delta=0.5$ provides a meaningful routing threshold:
> - If $\delta$ is too low, almost all pixels are considered uncertain, resulting in unnecessary refinement and loss of gating sparsity.
> - If $\delta$ is too high, only a few pixels are routed, making the refinement module underutilized.
>
> Thus, $\delta=0.5$ strikes a balance, activating residual experts in genuinely ambiguous regions while maintaining efficiency. Therefore, we adopt a fixed threshold $\delta=0.5$ for all five public datasets for consistency. We will include this explanation and the ablation results in the final version.
>
> **Q3&W3 “There is no study on the hyperparameters used in the gating module and residual experts, such as normalized routing weights {w_j}. A hyperparameter study would strengthen the contribution”**:
>
> Thank you for the helpful suggestion. We address the reviewer's concerns as follows:
>
> **1.** Routing weights $\{w_j\}$ are dynamically learned.
>
> The weights
> $
> \lbrace w_j \rbrace^J_{j=1}
> $
> are not manually defined hyperparameters. They are predicted dynamically by the Gating module $g_\theta$, which outputs both the uncertainty mask $m \in [0,1]^{H \times W}$ and expert routing weights $\{w_j\}$, based on input $x$ and coarse prediction $\hat{y}_{\mathrm{SAM}}$.
>
> **2.** Softmax normalization ensures stability.
>
> To enable robust and balanced routing, we apply softmax normalization to the predicted weights across experts at each spatial location $\sum_{j=1}^{J} w_j = 1$. This prevents expert collapse and allows smooth specialization without hard routing decisions.
>
> **3.** Explanation of the indicator function $\mathbb{I}(\cdot)$.
>
> The Gating module is trained using a binary supervision mask $g_t$ defined as:
>
> $$
> g_t(i) = \mathbb{I}\left(|\hat{y}_{\text{SAM}}(i) - y(i)| > \delta\right)
> $$
>
> where $\mathbb{I}(\cdot)$ denotes the standard indicator function, returning 1 if the condition holds and 0 otherwise. This provides supervision for pixels with high prediction error, helping the module identify uncertain regions.
>
> **4.** On the number of residual experts $J$.
>
> As shown in Tab.3 of our manuscript, we performed an ablation study on $J \in \{1, 2, 4, 6\}$. While $J=6$ achieved the highest accuracy, we chose $J=4$ in the final model to strike a better trade-off between segmentation performance and model complexity. This reduces the number of parameters and computational cost, while still maintaining strong performance.

---

> > ### Comment · Reviewer_6m4f · 2025-08-09
> >
> > After reviewing the rebuttal by the authors and other reviewers' comments, I keep my rating. The authors have addressed most of my concerns, and the SAM-based proposed method would be helpful in vessel segmentation masks of various medical images.

---

> > > ### Author Response · Authors · 2025-08-09
> > > **Author Response to Reviewer 6m4f comments**
> > >
> > > We thank the reviewer for their time and constructive feedback. We are pleased to hear that most of the reviewer’s concerns have been addressed and that they recognize the potential of our SAM-based method for vessel segmentation across various medical imaging modalities. We appreciate the reviewer’s consideration.

---

### Note · Authors · 2025-08-12

We thank all reviewers and the AC for constructive feedback. Most concerns were addressed in the rebuttal, with multiple reviewers expressing satisfaction and willingness to raise scores. Below is a per-reviewer summary of our responses.

### R6m4f

- **Parameter efficiency**: FineSAM++ adds only 1.4M params over SAM, outperforming other SAM-based methods in Avg. Dice.

- **Gating threshold $\delta$**: Ablation over $\{0.3,\dots,0.7\}$ shows $\delta=0.5$ optimal; same value used for all datasets.

- **Hyperparameters**: Routing weights learned dynamically; $J=4$ chosen for accuracy–efficiency balance; Clarified indicator function in gating supervision.

### RLmaj

- **Sparse related work & baselines**: Expanded Related Work; added HQ-SAM, nnUNet, RETFound to all datasets with unified augmentations.

- **Statistical robustness**: Reported mean $\pm$ std and paired T-tests; most improvements statistically significant, with lower variance.

- **Parameter efficiency**: FineSAM++ has 94.4M parameters (1.4M more than SAM’s 93.7M) and higher latency than UNet (34.53M), but achieves the highest Avg.Dice among all compared methods.

- **UNet variability**: Explained performance sensitivity to modality and contrast changes.

### RgHFg

- **Generalisability**: Extended evaluation to Synapse multi-organ CT, achieving highest mean Dice (87.97%) and lowest HD (7.89), confirming robustness beyond vessels.

- **Parameter efficiency**: Added comparison showing only 1.4M extra params over SAM, highest Avg.Dice among SAM-based methods.

- **Gating threshold $\delta$**: Ablation over $\{0.3,\dots,0.7\}$ shows $\delta=0.5$ optimal.

- **Method clarity**: Expanded $g_\theta$ and expert module descriptions for reproducibility.

- **Societal impact**: FineSAM++ improves clinical reliability via structure-aware, sparsely activated experts for fine-grained medical segmentation.

### RospL

- **Failure mitigation**: Added Degrade strategy (random corruption of coarse masks) to prevent thin-structure overfitting, improving Dice from 0.8154 to 0.8231.

- **Ablations**: Progressive uncertainty-guided optimization outperforms naïve joint (Dice 0.7984) and stage-wise (0.8117) training, achieving 0.8231 while avoiding overfitting.

- **Adapter comparison**: Showed SAM-Adapter underperforms for fine-grained vessels, supporting sparse MoE design.

These revisions enhance the quality of the paper, confirming FineSAM++ as an efficient method for fine-grained segmentation.

---

### Decision · Program_Chairs · 2025-09-17

**Decision:**

Accept (poster)

**Comment:**

The paper introduces FineSAM++, a structure-aware, sparse Mixture-of-Experts add-on for SAM that routes only uncertain, topology-critical regions (e.g., fine vessels) to lightweight Residual Experts for refinement, leaving confident regions untouched. The paper demonstrates a clear, substantive improvement on an important application with a simple, interpretable mechanism that is broadly useful for adapting foundation models in clinical settings. All the reviewers gave positive ratings. Therefore, meta reviewer agrees with the reviewers for the final recommendation.